# Improved Regret Bounds
# for Bandit Combinatorial Optimization[*]

**Shinji Ito**[†]
NEC Corporation, The University of Tokyo
i-shinji@nec.com

**Daisuke Hatano**
RIKEN AIP
daisuke.hatano@riken.jp

**Hanna Sumita**
Tokyo Metropolitan University
sumita@tmu.ac.jp

**Kei Takemura**
NEC Corporation
kei_takemura@nec.com

**Takuro Fukunaga**[‡]
Chuo University, RIKEN AIP, JST PRESTO
fukunaga.07s@g.chuo-u.ac.jp

**Naonori Kakimura**[§]
Keio University
kakimura@math.keio.ac.jp

**Ken-ichi Kawarabayashi**[§]
National Institute of Informatics
k-keniti@nii.ac.jp

## Abstract

*Bandit combinatorial optimization* is a bandit framework in which a player chooses an action within a given finite set $\mathcal{A} \subseteq \{0,1\}^d$ and incurs a loss that is the inner product of the chosen action and an unobservable loss vector in $\mathbb{R}^d$ in each round. In this paper, we aim to reveal the property, which makes the bandit combinatorial optimization hard. Recently, Cohen et al. [8] obtained a lower bound $\Omega(\sqrt{dk^3T/\log T})$ of the regret, where $k$ is the maximum $\ell_1$-norm of action vectors, and $T$ is the number of rounds. This lower bound was achieved by considering a continuous strongly-correlated distribution of losses. Our main contribution is that we managed to improve this bound by $\Omega(\sqrt{dk^3T})$ through applying a factor of $\sqrt{\log T}$, which can be done by means of strongly-correlated losses with *binary* values. The bound derives better regret bounds for three specific examples of the bandit combinatorial optimization: the multitask bandit, the bandit ranking and the multiple-play bandit. In particular, the bound obtained for the bandit ranking in the present study addresses an open problem raised in [8]. In addition, we demonstrate that the problem becomes easier without considering correlations among entries of loss vectors. In fact, if each entry of loss vectors is an independent random variable, then, one can achieve a regret of $\tilde{O}(\sqrt{dk^2T})$, which is $\sqrt{k}$ times smaller than the lower bound shown above. The observed results indicated that correlation among losses is the reason for observing a large regret.

[*]This work was supported by JST, ERATO, Grant Number JPMJER1201, Japan.
[†]This work was supported by JST, ACT-I, Grant Number JPMJPR18U5, Japan.
[‡]This work was supported by JST, PRESTO, Grant Number JPMJPR1759, Japan.
[§]This work was supported by JSPS, KAKENHI, Grant Number JP18H05291, Japan.

# 1  Introduction

This paper is aimed to investigate the *bandit combinatorial optimization* problem defined as follows: A player is given a finite *action set* $\mathcal{A} \subseteq \{a \in \{0,1\}^d \mid \|a\|_1 = k\}$ and the number $T$ of rounds for decision-making. In each round $t = 1, 2, \ldots, T$, the player chooses an *action* $a_t$ from $\mathcal{A}$. At the same time, the environment privately chooses a *loss vector* $\ell_t = [\ell_{t1}, \ldots, \ell_{td}]^\top \in [0,1]^d$, and the player observes the loss $\ell_t^\top a_t$ incurred by the action $a_t$. The goal of the player is to minimize the expected cumulative loss $\mathbf{E}[\sum_{t=1}^T \ell_t^\top a_t]$, where the expectation is taken with respect to the player's internal randomization. The performance of the algorithm is measured in terms of the *regret* $R_T$ defined by $R_T = \max_{a \in \mathcal{A}} \mathbf{E}\left[ \sum_{t=1}^T \ell_t^\top a_t - \sum_{t=1}^T \ell_t^\top a \right]$.

In this study, we focus on the minimax regret, the worst-case regret attained by optimal algorithms, which can be expressed as $\mathcal{R}_T := \min_{\text{algorithm}} \max_{\{\ell_t\}_{t=1}^T \subseteq [0,1]^d} R_T$. The minimax regret can be bounded from above by designing algorithms. The current best bound is $\mathcal{R}_T = O(\sqrt{dk^3 T \log(ed/k)})$, as reported in a number of papers [2; 6; 7; 10; 12]. However, lower bounds of the minimax regret can be proven by constructing a probabilistic distribution of loss vectors for which any algorithm incurs a certain degree of regret. To obtain a lower bound, Audibert et al. [2] constructed a probabilistic distribution of loss vectors for which arbitrary algorithms incurred a regret of $\Omega(\sqrt{dk^2 T})$, and they conjectured that this bound was tight, i.e., $\mathcal{R}_T = \Theta(\sqrt{dk^2 T})$. Recently, however, Cohen et al. [8] presented the lower bound of $\mathcal{R}_T = \Omega(\sqrt{dk^3 T / \log T})$, which rejected the above-mentioned conjecture, and thereby, they have decreased the gap between the upper and lower bounds to $O(\sqrt{\log(ed/k) \log T})$ consisting of logarithmic terms only.

The input distribution constructed by Cohen et al. [8] to derive the lower bound has the unique characteristics that cannot be found in previous studies, such as lower bounds for a multi-armed bandit [4], a combinatorial semi-bandit [6; 20; 23] and a combinatorial bandit [2]. In previous studies on lower bounds, only binary inputs and an arm-wise independent distribution were considered, i.e., $\ell_{t1}, \ldots, \ell_{td}$ are mutually independent $\{0,1\}$-valued discrete random variables. Such inputs were proved to result in tight lower bounds for multi-armed bandits [4] and combinatorial semi-bandits [2; 20]. In contrast to these studies, Cohen et al. [8] introduced loss vectors following a continuous distribution over $[0,1]^d$ and having a strong correlation among $d$ entries. Furthermore, the lower bound obtained in Cohen et al. [8] includes a $1/\sqrt{\log T}$ term, which does not appear in the other lower bounds for bandit problems. In addition, they applied the obtained lower bounds to special cases, such as the multitask bandit and the bandit ranking problem. However, their results are restricted to the problems under certain parameter constraints, and consequently, the task of identifying the tight bounds for some important special cases, including the problem referred to as *bandit ranking with full permutations*, were left open.

Such characteristics corresponding to the input distribution defined by Cohen et al. [8] lead to the following research questions:

**Q. 1** Is the $1/\sqrt{\log T}$ factor in the lower bound given by Cohen et al. [8] redundant or inevitable?

**Q. 2** Does the continuous distribution of loss vectors make the problem essentially harder than the discrete (binary) distribution? If we restrict our consideration to the loss vectors in $\{0,1\}^d$, then the player can see the *number of good arms* ($i \in [d]$ s.t. $l_{ti} = 0$) in the chosen arms $S_t$, which may, or may not, be more informative than actual values.

**Q. 3** Does the correlation of loss among different arms make the problem essentially harder than the arm-wise independent loss?

**Q. 4** Can we obtain tight lower bounds for the special cases such as the bandit ranking problem with full permutations resolving the open question in [8]?

# 2  Main Results

Our main results can be interpreted to answer the above four questions. First, we improve the regret lower bound obtained by [8] to $\Omega(\sqrt{dk^3 T})$, by applying a factor of $\sqrt{\log T}$, as shown in Table 1. These bounds can be proven by constructing a distribution of *strongly-correlated* losses using *binary* values. We apply the bounds to the three specific examples of bandit combinatorial optimization

Table 1: Regret bounds $\mathcal{R}_T$ for bandit combinatorial optimization.

| Assumption | Upper bound by Algorithms | Lower bound |
|---|---|---|
| No assumption | $O(\sqrt{dk^2T\log\|\mathcal{A}\|})$ $= O(\sqrt{dk^3T\log(ed/k)})$ ([6] and [7]) | $\Omega(\sqrt{dk^3T/\log T})$ by $\ell_t \in [0,1]^d$ ([8]), $\Omega(\sqrt{dk^3T})$ by $\ell_t \in \{0,1\}^d$ (**Theorems 1** and **2**) |
| Independent losses | $O(\sqrt{dkT\log\|\mathcal{A}\|\log T})$ $= O(\sqrt{dk^2T\log(ed/k)\log T})$ (**Algorithm 1** and **Theorem 3**) | $\Omega(\sqrt{dk^2T})$ by $\ell_t \in \{0,1\}^d$ ([2]) |

that have high practical importance: the *multitask bandit problem*, the *bandit ranking problem* (Theorem 1), and the *multiple-play bandit problem* (Theorem 2). This result provides answers to **Q. 1** and **Q. 2** outlined in Section 1: The $1/\sqrt{\log T}$ factor in the lower bound is redundant, and the difference between continuous-valued and discrete-valued losses does not have a large impact on the hardness of the problem. This observation also addresses **Q. 4**, an open problem outlined in [8].

The multitask bandit problem [7; 8] is a bandit framework in which the player tries to solve $k$ instances of the $n$-armed bandit problem. This is a special case of the bandit combinatorial optimization with $d = kn$ and

$$\mathcal{A} = \left\{ a \in \{0,1\}^d \ \middle| \ \sum_{i=(j-1)n+1}^{jn} a_i = 1 \quad (j \in [k]) \right\}. \tag{1}$$

In the bandit ranking problem or online ranking problem [13] with bandit feedback problem, the goal of the player is to find a maximum matching in the complete bipartite graph $K_{k,n}$ with $d = kn$ edges, where $k \in [n]$. The set of all maximum matchings can be expressed as follows:

$$\mathcal{A} = \left\{ a \in \{0,1\}^d \ \middle| \ \sum_{i=(j-1)n+1}^{jn} a_i = 1 \ (j \in [k]), \ \sum_{i=1}^{k} a_{(i-1)n+j} \leq 1 \ (j \in [n]) \right\}. \tag{2}$$

Considering these problems, we obtain the following regret lower bound.

**Theorem 1** (multitask bandit, bandit ranking). *Suppose that $\mathcal{A}$ is defined by* (1) *or* (2) *and $n \geq 2$. There is a probability distribution $D$ over $\{0,1\}^d$ for which the following statement holds: If $\ell_t$ is drawn from $D$ for $t = 1, \dots, T$ independently, the regret for any algorithm satisfies $\mathbf{E}[R_T] = \Omega(\min\{\sqrt{dk^3T}, k^{3/4}T\})$, where the expectation is taken with respect to the randomness of $\ell_t$.*

Considering the bandit ranking problem, the previous work [8] demonstrated the lower bound of $\Omega(\sqrt{dk^3T/\log T})$ under the assumption of $n \geq 2k$, and the full-permutation case ($k = n$) was left as an open problem, as mentioned in the conclusion of this research work. Theorem 1 answers to this open problem: Even if $k = n$, the minimax regret is of $\mathcal{R}_T = \tilde{\Theta}(\sqrt{dk^3T}) = \tilde{\Theta}(\sqrt{k^5T})$, ignoring a $\sqrt{\log k}$ factor. Theorem 1 can also be extended to the online shortest path problem [5], by the standard reduction from multitask bandit to the online shortest path. See e.g., [8] for details of the reduction.

The multiple-play bandit problem [7; 16; 18; 23] is another bandit framework in which the player can choose arbitrary $k$ arms from a set of $d$ arms in each round. This problem corresponds to $\mathcal{A} = \binom{[d]}{k} := \{a \in \{0,1\}^d \mid \|a\|_1 = k\}$.

**Theorem 2** (multiple-play bandit). *Suppose that $\mathcal{A} = \binom{[d]}{k}$. There is a probability distribution $D$ over $\{0,1\}^d$ for which the following holds: If $\ell_t$ is drawn from $D$ for $t = 1, \dots, T$ independently, the regret for any algorithm will satisfy $\mathbf{E}[R_T] = \Omega\left(\min\left\{(\frac{d-k}{d})^2\sqrt{dk^3T}, \frac{d-k}{d}k^{3/4}T\right\}\right)$, where the expectation is taken with respect to the randomness of $\ell_t$.*

The above lower bound means that $\mathcal{R}_T = \Omega(\sqrt{dk^3T})$ for $T = \Omega(dk^{3/2})$ and $d = \Omega(k)$. It should be noted that existing works [2; 8; 20] provided weaker lower bounds only for the case of $d \geq 2k$,

while those provided in the present study are valid for general $d$ and $k$. The proof of Theorem 2 is presented in Appendix C.

A basic idea for proving a nearly tight bound is to construct an environment, where all entries of $\ell_t$ are strongly correlated between each other; this concept has been introduced by Cohen et al. [8]. If losses are strongly correlated, the observed value $\ell_t^\top a$ has a larger variance. For example, the variance is of order $k$ if all entries are independent, while it can be of order $k^2$ if all entries take the same value. When the observed values $\ell_t^\top a$ have larger variance, the KL divergence among the values for different actions $a$ is small, which implies that no algorithm can detect "good" actions properly. Cohen et al. [8] constructed such an environment by means of normal distributions, which improve the lower bound by $\tilde{O}(\sqrt{k})$ factor. However, their proposed bound includes a redundant $(\log T)^{-1/2}$ factor due to the unbounded support of normal distributions.[1] We note that their technique has been used recently for proving a lower bound for *bandit PCA* [17], which includes a redundant $(\log T)^{-1/2}$ factor too, for the same reason as the above.

To shave off the $(\log T)^{-1/2}$ factor, in this paper, we introduce a novel class of discrete distributions over $\{0, 1\}^d$, so that entries of loss vectors are bounded and strongly correlated. To make the losses correlated, we consider $d$ Bernoulli distributions that share the parameter, by which the observed value has a large variance of $O(k^2)$. However, it is not a straightforward task to set "good" actions in this approach. The previous work [8] simply decreases the mean parameter in the normal distribution to set "good" actions, but it does not work as it causes large KL divergences between "good" actions and the others in our distribution. In the present work, we adjust the parameter of Bernoulli distributions carefully with the intention of ensuring small KL divergence, which allows improving the regret lower bound successfully. The idea outlined in the present study can be used to improve the idea of [8] even considering other problems.

Second, we show that the correlation among losses is the reason of observing a large regret. In fact, if each entry of loss vectors is an independent random variable, then one can achieve a regret of $\tilde{O}(\sqrt{dk^2T})$ as below, which is $\sqrt{k}$ times smaller than the lower bounds in Theorems 1 and 2. This provides the answer to **Q. 3**: The correlation among losses makes the problem essentially harder, as the minimax regret bound becomes larger by a factor of $\tilde{\Theta}(\sqrt{k})$.

**Theorem 3** (smaller regret bound for the arm-wise independent loss)**.** *There exists an algorithm that achieves* $\mathbf{E}[R_T] = O(\sqrt{dk^2T \log T \log(ed/k)})$ *for* $T = \Omega(d^3)$*, under the assumption that* $\ell_t$ *follows a distribution of mutually independent $d$ random variables in* $[0, 1]$*, i.i.d. for* $t = 1, 2, \dots, T$*.*

This upper bound is nearly tight; Theorem 5 in [2] implies that any algorithm suffers $\mathbf{E}[R_T] = \Omega(\sqrt{dk^2T})$ in the worst case under the same assumption as in Theorem 3.[2] By combining this result and Theorem 3, we obtain the following corollary:

**Corollary 1.** *Under the same assumption as in Theorem 3, the minimax regret in the bandit combinatorial optimization is of order* $\tilde{\Theta}(\sqrt{dk^2T})$*, where we ignore logarithmic factors in $d$ and $T$.*

To prove Theorem 3, we analyze regret upper bounds for *stochastic linear bandits*, which are generalization of the bandit combinatorial optimization with stochastic environments. In stochastic linear bandits, a player is given a finite set $\mathcal{A} \subseteq \mathbb{R}^d$ of $d$-dimensional vectors. In each round, the player chooses $a_t \in \mathcal{A}$ and receives loss $L_t = \ell^{*\top} a_t + \eta_t$, where $\eta_t$ is the noise, which is assumed to be conditionally $\alpha$-subgaussian. We also assume that $\sup_{a,b \in \mathcal{A}} \ell^{*\top}(a - b) \leq L$. We observe that bandit combinatorial optimization with the assumption defined in Theorem 3 is a special case of stochastic linear bandits with $\alpha = \sqrt{k}/2$ and $L = k$.

For stochastic linear bandits with $\alpha = 1$ and $L = 1$, Lattimore and Szepesvári [19] provided an algorithm that achieves $R_T = O(\sqrt{dT \log \frac{|\mathcal{A}| \log T}{\delta}})$.[3] This upper bound, however, does not directly lead to Theorem 3, because their bound holds only for the case of $\alpha = 1$ and $L = 1$; If we directly

apply their result, we obtain $R_T = O(\sqrt{dk^2 T \log \frac{|\mathcal{A}| \log T}{\delta}}) = \tilde{O}(\sqrt{dk^3 T})$ by multiplying losses by $1/k$. This is $\tilde{\Omega}(\sqrt{k})$ times larger than the bound provided in Theorem 3.

To mitigate this issue, we modify their algorithm, so that we can perform a more refined analysis for the case of arbitrary $\alpha$ and $L$. The differences between our Algorithm 1 given in Appendix D.1 and Algorithm 12 in [19] are summarized as follows:

- They deal with only the case in which the noise $\eta_t$ has a bounded variance, i.e., $\alpha = 1$. To deal with the case for a general $\alpha$, we modify the definition (31) of $T_k$ in their algorithm.
- They assume that the suboptimality gap $\max_{a,b \in \mathcal{A}}\{\ell^{*\top}(a - b)\}$ is bounded by 1. To handle properly the changing suboptimality gaps, we modify the definition of $\varepsilon_t$ in their algorithm.
- They basically consider maximization problems, while we consider minimization (This doe not result in essential differences).

We demonstrate that Algorithm 1 achieves the following regret bound:

**Theorem 4.** *For any input parameters $\delta > 0$ and $\varepsilon_1 > 0$, with a probability of at least $1 - \delta$, the output of Algorithm 1 satisfies*

$$\max_{a \in \mathcal{A}} \sum_{t=1}^{T} \ell^{*\top}(a_t - a) \leq 9\alpha \sqrt{dT \log \frac{|\mathcal{A}| \log T}{\delta}} + L \frac{2d\alpha^2}{\varepsilon_1^2} \log \frac{2|\mathcal{A}|}{\delta} + (L + \varepsilon_1)d^2. \quad (3)$$

Theorem 4 means that the upper bound $L$ of $\ell^{*\top} a_t$ does not affect the leading term of the regret upper bound, however, $\alpha$ does affect. By substituting $\alpha = \sqrt{k}/2$ and $L = k$ with the bound in Theorem 4, we obtain Theorem 3.

## 3  Related Work

Bandit combinatorial optimization was first introduced by McMahan and Blum [21] and Awerbuch and Kleinberg [5]. They proposed the algorithms achieving the regret of $\tilde{O}(T^{3/4})$ and $\tilde{O}(T^{2/3})$, respectively, ignoring dependence on $d$ and logarithmic factors in $T$. Algorithms with improved regret bounds have been proposed in several papers [2; 6; 7; 10]. These algorithms achieve $R_T = O(\sqrt{dk^3 T \log(ed/k)})$ in our problem setting. Recently, computationally efficient algorithms achieving the sublinear regret have also been introduced in [7; 9; 12; 22; 14].

With regard to lower bounds in the bandit combinatorial optimization, Audibert et al. [2] showed that $R_T = \Omega(\sqrt{dk^2 T})$, and consequently, they conjectured that this lower bound was tight. However, the recent work by Cohen et al. [8], rejected this conjecture showing that $R_T = \Omega(\sqrt{dk^3 T / \log T})$.

Combinatorial semi-bandit optimization is a variant of bandit combinatorial optimization, in which the player can observe not only the total loss $\ell_t^\top a_t$, but also the entry $\ell_{ti}$ for each chosen arm $i \in S_t$. This problem was introduced by György et al. [11] in the context of the online shortest path problem, i.e., they considered the case in which $\mathcal{A}$ is a set of all subsets of edges constructing a path in a given graph. For general action sets $\mathcal{A} \subseteq \binom{[d]}{k}$, Audibert et al. [2] proposed an algorithm achieving the regret of $O(\sqrt{dkT})$, and showed that it is minimax optimal, i.e., there is an action set $\mathcal{A} \subseteq \binom{[d]}{k}$ such that $R_T = \Omega(\sqrt{dkT})$. With regard to the multiple-play bandit problem, i.e., the case of $\mathcal{A} = \binom{[d]}{k}$, with semi-bandit feedback, Uchiya et al. [23] showed that $R_T = \Omega(\sqrt{dT})$, but it remained open whether this bound was tight, until the recent work by Lattimore et al. [20] provided the proof that $R_T = \Omega(\sqrt{dkT})$.

The study on stochastic linear bandits was introduced in the work by Abe and Long [1]. They and Auer [3] considered the case of finite action sets that can change every round. Bandit combinatorial optimization with a stochastic environment can be seen as a special case of stochastic linear bandits in which the action set is included in $\binom{[d]}{k}$ and does not change in every round. Auer [3] introduced a technique of dividing rounds to achieve $R_T = O(\sqrt{dT(\log(|\mathcal{A}|T \log T))^3})$ under the assumption of bounded loss. We remark that a similar technique is used in Algorithm 1. Moreover, a similar technique was used for spectral bandits considered by Valko et al. [24], in which they eliminated inappropriate arms over several phases.

# 4 Lower Bounds

In this section, we provide proofs for Theorems 1 and 2. First, we revisit the proofs presented in the previous work: Theorem 5 in [2] and Lemma 4 in [8], which provide the regret lower bounds of the order $\Omega(\sqrt{dk^2T})$ and $\Omega(\sqrt{dk^3T/\log T})$ for multitask bandits, respectively. From the proofs provided in the related work, we can observe that regret lower bounds can be derived from upper bounds on KL divergences determined by distributions of loss vectors. Second, we construct a distribution of loss vectors, so that the corresponding KL divergence is small enough. Combining these two results, we obtain Theorem 1, which provides an improved lower bound for multitask bandits. Finally, we extend the proof for multitask bandit to prove Theorem 2 for multiple-play bandits.

## 4.1 Proof idea used in the previous work

This subsection revisits the proofs for regret lower bounds for multitask bandit, given in [2] and [8]. We note that, from Yao's minimax principle, it suffices to construct a probabilistic distribution of $\ell_t$, such that in expectations, any deterministic algorithm suffers large regret.

In both proofs, the probabilistic distribution of the loss vectors is defined as follows. First, it is necessary to set a parameter $\varepsilon > 0$, which is to be optimized later. For $a^* = [a_1^*, \ldots, a_d^*]^\top \in \{0,1\}^d$, a probabilistic distribution $D_{a^*}$ over $\mathbb{R}^d$ is defined such that $\ell \sim D_{a^*}$ satisfies

$$\mathop{\mathbf{E}}_{\ell \sim D_{a^*}}[\ell_i] = \frac{1}{2} - \varepsilon a_i^* \tag{4}$$

for each $i \in [d]$. More concretely, [2] define $D_{a^*}$ such that the $i$-th entry of the vector follows the Bernoulli distribution of parameter $\frac{1}{2} - \varepsilon a_i^*$, independently. Cohen et al. [8] define $D_{a^*}$ such that the $i$-th entry is equal to $\frac{1}{2} - \varepsilon a_i^* + Z$, where $Z$ follows the normal distribution $N(0, \sigma^2)$. We can confirm that these two definitions satisfy (4). The environment picks $a^* \in \mathcal{A}$ uniformly at random before the game begins, and then, in round $t = 1, 2, \ldots, T$, generates a loss vector $\ell_t$ following $D_{a^*}$ i.i.d. It should be noted that $\mathcal{A}$ is defined by (1) here.

We analyze the regret bounds for these loss vectors. Let $S^* = \{i \in [d] \mid a_i^* = 1\}$, and $a_t$ be the action chosen by the player in round $t$. Let us define $N_i$ to be the number of rounds in $[T]$ in which the player suffers a loss for the $i$-th entry of loss vectors, i.e., $N_i = |\{t \in [T] \mid a_{ti} = 1\}|$. Then, from (4), the regret $R_T$ satisfies

$$\mathop{\mathbf{E}}_{\ell_1,\ldots,\ell_T \sim D_{a^*}}[R_T] \geq \mathop{\mathbf{E}}_{\ell_1,\ldots,\ell_T \sim D_{a^*}}\left[\sum_{t=1}^{T} \ell_t^\top a_t - \sum_{t=1}^{T} \ell_t^\top a^*\right] = \varepsilon\left(kT - \sum_{i \in S^*} \mathop{\mathbf{E}}_{\ell_1,\ldots,\ell_T \sim D_{a^*}}[N_i]\right). \tag{5}$$

From (5), to obtain a lower bound on $R_T$, it suffices to bound $\sum_{i \in S^*} N_i$. To obtain a bound on $N_i$, we use the following lemma:

**Lemma 1.** *Let $D$ and $D'$ be the probability distributions over $[0,1]^d$. Then, we have*

$$\left|\mathop{\mathbf{E}}_{\ell_1,\ldots,\ell_T \sim D}[N_i] - \mathop{\mathbf{E}}_{\ell_1,\ldots,\ell_T \sim D'}[N_i]\right| \leq T\sqrt{\sum_{t=1}^{T} \mathop{\mathbf{E}}_{a_t \sim A_t(D)}\left[\mathop{\mathbf{KL}}_{\ell \sim D, \ell' \sim D'}(a_t^\top \ell \| a_t^\top \ell')\right]} \tag{6}$$

*for any deterministic algorithm, where $A_t(D)$ represents the probability distribution of the outputs of the algorithm in round $t$ for the inputs $\ell_1, \ell_2, \ldots, \ell_{t-1}$ following $D$ independently.*

This lemma follows from Pinsker's inequality and the chain rule for the KL divergence. For details, see, e.g., Lemma A.1. in [4].

Lemma 1 defines a connection between bounds on $N_i$ and upper bounds on KL divergences of specific distributions. To provide a bound on $N_i$ by means of Lemma 1, Audibert et al. [2] and Cohen et al. [8] used specific properties of their distributions. We observe that their arguments are focused on the fact that their distributions of loss vectors satisfy the following condition regarding the KL divergence:

$$a \in \mathcal{A}, \ a^*, \hat{a} \in \{0,1\}^d, \ \hat{a}^\top a - a^{*\top} a = 1, \ \ell^* \sim D_{a^*}, \hat{\ell} \sim D_{\hat{a}}$$

$$\implies \quad \mathbf{KL}(\ell^{*\top} a \| \hat{\ell}^\top a) \leq C_D \varepsilon^2 \text{ for a constant } C_D \text{ depending on } \{D_a\}. \tag{7}$$

Intuitively, the precondition of (7) means that the discrepancy with respect to the expected loss is at most $\varepsilon$. In fact, $a^{*\top}a$ in (7) corresponds to "goodness of action $a$" for the loss vector $\ell \sim D_{a^*}$, because the expected loss for action $a$ is equal to $k/2 - \varepsilon a^{*\top}a$ from (4). Consequently, $\hat{a}^{\top}a - a^{*\top}a = 1$ means that the expected loss for $D_{a^*}$ is smaller than one for $D_{\hat{a}}$ by $\varepsilon$.

We can show that, if (7) is true, then Lemma 1 implies that if $a^*$ follows a uniform distribution over $\mathcal{A}$ defined by (1), we obtain $\mathop{\mathbf{E}}\limits_{a^*,\ell_1,\dots,\ell_T \sim D_{a^*}}\left[\sum_{i \in S^*} N_i\right] \leq k\left(\frac{T}{2} + T\varepsilon\sqrt{\frac{kT}{d}C_D}\right)$. Therefore, if we set $\varepsilon \leq \sqrt{d/(16C_DkT)}$, we obtain $\mathop{\mathbf{E}}\limits_{a^*,\ell_1,\dots,\ell_T \sim D_{a^*}}\left[\sum_{i \in S^*} N_i\right] \leq 3kT/4$, and consequently, we obtain $\mathbf{E}[R_T] \geq \frac{\varepsilon kT}{4}$ from (5). The main observation of this subsection is summarized as follows:

**Observation 1.** *Suppose a family $\{D_{a^*} \mid a^* \in \{0,1\}^d\}$ of distributions with a parameter $\varepsilon \leq \sqrt{d/(16C_DkT)}$ that satisfies (4) and (7). If $\mathcal{A}$ is given by (1) with $n \geq 2$, then we have a regret lower bound of $\mathbf{E}[R_T] = \Omega(\varepsilon kT)$.*

### 4.2 Construction of the probabilistic distribution

The goal of this subsection is to construct a family $\{D_{a^*} \mid a^* \in \{0,1\}^d\}$ of distributions such that (4) and (7) are satisfied with $C_D = O(1/k^2)$. From Observation 1, such construction leads to a regret lower bound of $\mathbf{E}[R_T] = \Omega(\sqrt{dk^3T})$ for the multitask bandit problem, thereby, proving Theorem 1.

The proposed probabilistic distribution of loss vectors is defined as follows. Let us set a parameter $\varepsilon \in [0, 2^{-16}]$, which is to be optimized later. For $a^* = [a_1^*, \dots, a_d^*]^{\top} \in \{0,1\}^d$, let $D_{a^*}$ be a distribution of $\ell = [\ell_1, \dots, \ell_d]^{\top} \in [0,1]^d$ generated in the following way:

(i) Draw $u_0$ from a uniform distribution over $[0,1]$. $\hspace{2cm}$ (8)

(ii) Draw $b_i$ from a Bernoulli distribution of parameter $(1/2 + 2\varepsilon a_i^*)$.

(iii) For $i \in [d]$, draw $u_i$ from a uniform distribution over $\begin{cases} [0, 1/2] & \text{if } b_i = 1, \\ (1/2, 1] & \text{if } b_i = 0. \end{cases}$

(iv) Let $\ell_i = 1$ if $u_i \geq u_0$, and otherwise, $\ell_i = 0$.

We can confirm that (4) holds for this $D_{a^*}$. In fact, step (iv) means $\mathbf{E}[\ell_i] = \mathrm{Prob}[u_i \geq u_0]$, and as $u_0$ follows the uniform distribution over $[0,1]$ and $u_i \in [0,1]$, we obtain $\mathrm{Prob}[u_i \geq u_0] = \mathbf{E}[u_i]$. Moreover, from steps (ii) and (iii), we obtain $\mathbf{E}[u_i] = \frac{1}{4}\mathrm{Prob}[b_i = 1] + \frac{3}{4}\mathrm{Prob}[b_i = 0] = \frac{1}{4}(\frac{1}{2} + 2\varepsilon a_i^*) + \frac{3}{4}(\frac{1}{2} - 2\varepsilon a_i^*) = \frac{1}{2} - \varepsilon a_i^*$, which means that (4) holds.

Let us show that (7) is satisfied with $C_D = O(1/k^2)$. As $D_{a^*}$ is a distribution over $\{0,1\}^d$, $\ell^{*\top}a$ takes values from $\{0,1,\dots,k\}$ for any $a \in \mathcal{A}$ and $\ell^* \sim D_{a^*}$. For $i = 0,1,\dots,k$, define $P(i) = \mathrm{Prob}[\ell^{*\top}a = i]$ and $P'(i) = \mathrm{Prob}[\hat{\ell}^{\top}a = i]$, where $\ell^* \sim D_{a^*}$ and $\hat{\ell} \sim D_{\hat{a}}$. Then, from the definition, the KL divergence can be expressed as follows:

$$\mathbf{KL}(\ell^{*\top}a \| \hat{\ell}^{\top}a) = -\sum_{i=0}^{k} P(i) \log \frac{P'(i)}{P(i)} = -\sum_{i=0}^{k} P(i) \log\left(1 + \frac{P'(i) - P(i)}{P(i)}\right)$$

$$\leq -\sum_{i=0}^{k} P(i)\left(\frac{P'(i) - P(i)}{P(i)} - 2\left(\frac{P'(i) - P(i)}{P(i)}\right)^2\right) = 2\sum_{i=0}^{k} \frac{(P'(i) - P(i))^2}{P(i)},$$

where the inequality comes from the fact that $\log(1 + x) \geq x - 2x^2$ for $|x| \leq 1/2$ and $|P'(i) - P(i)|/P(i) \leq 1/2$ holds,[4] and the last equality holds, as we have $\sum_{i=0}^{k} P(i)\frac{P'(i) - P(i)}{P(i)} = \sum_{i=0}^{k}(P'(i) - P(i)) = 1 - 1 = 0$. Thereby, it suffices to bound $(P'(i) - P(i))^2/P(i)$ for deriving an upper bound on the KL divergence. We can then show that $P(i) = \Omega(1/k)$ for all $i$. Indeed, if $\varepsilon = 0$, then we have $P(i) = 1/(k+1)$; as $\mathrm{Prob}[\ell_{ti} = 1] = \mathrm{Prob}[u_i \geq u_0]$ from the definition (8) of $D_a$, and as each $u_i$ is a uniform random variable over $[0,1]$ under the condition of $\varepsilon = 0$, we have

$$P(i) = \mathrm{Prob}\left[\sum_{j=1}^{k} \ell_{tj} = i\right] = \mathrm{Prob}\left[u_0 \text{ is the } (i+1)\text{-th smallest among } \{u_j\}_{j=0}^{k}\right] = \frac{1}{k+1},$$

where the last equality comes from the fact that $u_0, u_1, \ldots, u_k$ are i.i.d. random variables. Even if $\varepsilon > 0$, we show in Appendix A that for $\varepsilon \le 2^{-16}$, $P(i)$ is sufficiently close to $\frac{1}{k+1}$ to have an order of $\Omega(1/k)$. Thus, we have $P(i) = \Omega(1/k)$ for all $i = 1, \ldots, k$ and $\varepsilon \in [0, 2^{-16}]$, and hence, $\mathbf{KL}(\ell^{*\top} a || \hat{\ell}^\top a) = O\left(k \sum_{i=0}^{k} (P'(i) - P(i))^2\right)$. Finally, by proving $|P'(i) - P(i)| = O(\varepsilon/k^2)$, we obtain the following lemma:

**Lemma 2.** *Let* $a^*, \hat{a} \in \{0,1\}^d$ *and* $\ell^* \sim D_{a^*}, \hat{\ell} \sim D_{\hat{a}}$. *Then, for* $\varepsilon \in [0, 2^{-16}]$ *and* $a \in \{0,1\}^d$ *satisfying* $\|a\|_1 = k$ *and* $\hat{a}^\top a - \hat{a}^{*\top} a = 1$, *we have*

$$\mathbf{KL}(\ell^{*\top} a || \hat{\ell}^\top a) = O\left(\frac{\varepsilon^2}{k^2} + \frac{\varepsilon^4}{k^{3/2}}\right). \tag{9}$$

The complete proof of this lemma is provided in Appendix A.

### 4.3 Improved lower bound for the multitask bandit problem

We obtain an improved lower bound for $\mathcal{A}$ defined as (1), by combining Observation 1 and Lemma 2. From Lemma 2, if $\varepsilon \le 2^{-16} k^{-\frac{1}{4}}$, there is a global constant $C$ for which (7) holds with $C_D = (C/k)^2$. Consequently, setting $\varepsilon = \min\{2^{-16} k^{-\frac{1}{4}}, \frac{1}{4C}\sqrt{\frac{dk}{T}}\}$, we obtain $\mathbf{E}[R_T] = \Omega(\varepsilon k T) = \Omega(\min\{k^{\frac{3}{4}} T, \sqrt{dk^3 T}\})$, which provides the lower bound in Theorem 1, for $\mathcal{A}$ given by (1), i.e., the multitask bandit problem. The key point for shaving off the $\sqrt{\log T}$ factor is that our probabilistic distribution presented in Section 4.2 satisfies (7) with $C_D = O(1/k^2)$, while the previous work [8] does not exceed $C_D = O(\log T/k^2)$.

### 4.4 Improved and extended lower bound for the bandit ranking problem

For the bandit ranking problem, Cohen et al. [8] have identified lower bounds by considering $\ell_t \sim D_{a^*}$ for $a^* \in \mathcal{A}$, similar to the multitask bandit problem. However, this approach does not work well for the case of full permutations (i.e., with $k = n$), and has left an $\Omega(\sqrt{n})$-gap between the lower and the upper bounds, as mentioned in the conclusion of this research work.

We can eliminate this $\Omega(\sqrt{n})$-gap by improving the lower bound by a surprisingly simple approach. In contrast to the probability distribution considered by Cohen et al. [8] that has $k$ good arms ($i$ such that $a_i^* = 1$), we define the probability distribution with $m = \lceil k/2 \rceil$ good arms, i.e., we consider $a^* \in \mathcal{A}' \subseteq \{0,1\}^d$ defined by

$$\mathcal{A}' = \left\{ a \in \{0,1\}^d \;\middle|\; \sum_{i=(j-1)n+1}^{jn} a_i = \left\{ \begin{array}{ll} 1 & (1 \le j \le m) \\ 0 & (m < j \le k) \end{array} \right., \; \sum_{i=1}^{k} a_{(i-1)n+j} \le 1 \quad (j \in [n]) \right\}. \tag{10}$$

**Lemma 3.** *Suppose a family* $\{D_{a^*} \mid a^* \in \{0,1\}^d\}$ *of distributions with a parameter* $\varepsilon \le \sqrt{d/(32 C_D k T)}$ *that satisfies* (4) *and* (7). *Suppose* $n \ge 2$ *and* $1 \le k \le n$. *If* $a^*$ *is chosen from* $\mathcal{A}'$ *defined by* (10), *and* $\ell_t$ *follows* $D_{a^*}$ *for* $t = 1, 2, \ldots, T$, *independently, then, for the bandit ranking problem defined by* (2), *any algorithm suffers regret of* $\mathbf{E}[R_T] = \Omega(\varepsilon k T)$.

The proof of this lemma is provided in Appendix B.

The lower bound in Lemma 3 is valid even if $k = n$, while the approach of the previous work [8] considering $a^* \in \mathcal{A}$ is applicable only to the case of $n \ge 2k$. Intuitively, this difference can be explained as follows: the regret depends on the number of good arms ($i \in [d]$ such that $a_i^* = 1$) in chosen arms ($i \in [d]$ such that $a_{ti} = 1$). If $a^*$ and $a_t$ are chosen from $\mathcal{A}$ with $k = n$, and if the chosen arms (defined by $a_t$) include $k - 1$ good arms, then the chosen arms automatically include the entire $[k]$ good arms, because $a^*$ and $a_t$ express edge sets of perfect matchings of the complete bipartite graph $K_{k,k}$. This means that, in this setting, the probability of choosing several good arms strongly affects that of choosing other good arms, which makes the analysis difficult. However, such an effect can be reduced if $a^*$ is chosen from $\mathcal{A}'$, i.e., $a^*$ has only $m = \lceil k/2 \rceil$ good arms.

The lower bound in Theorem 1 for the bandit ranking problem, i.e., $\mathcal{A}$ given by (2), can be derived in the same way as in Section 4.3. This accomplishes the proof of Theorem 1.

# 5  Conclusion

In this study, we considered the regret bounds of the bandit combinatorial optimization. As a result, we managed to improve the regret lower regret bounds comparing with those presented in the existing study [8] by applying a factor of $\sqrt{\log T}$. The obtained lower bounds apply to three practically important examples of the bandit combinatorial optimization, and are valid under the parameter constraints milder than those outlined in the existing studies. In particular, the bound for the bandit ranking obtained in the present study addresses an open problem outlined in [8]. To shave off $\sqrt{\log T}$ factor, we have introduced a novel class of distributions, which could be potentially used to improve regret lower bounds considering other problems. Moreover, by obtaining a lower regret bound under the assumption of independent losses, we demonstrated that correlation among losses is the cause of observing a large regret.

With respect to the bandit combinatorial optimization, we decreased the gap between the upper and the lower bounds to $O(\log(ed/k))$. We will consider this issue as an open question for the future research, in which we will aim to improve the gap to a constant factor only.

## Footnotes

[1] To keep $\ell_t$ in the bounded region $[0, 1]^d$ with high probability, the variances of normal distributions need to be maintained sufficiently small, which makes the KL divergence large.

[2] Although the original statement in [2] does not include the independence assumption, we can confirm that it is satisfied in their proof.

[3] In their book, the proof is left for the reader as an exercise.

[4] The statement $|P'(i) - P(i)|/P(i) \leq 1/2$ comes from $\varepsilon \leq 2^{-16}$. See Appendix A for details.

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
