[Supplementary Material · main.pdf]

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

# A  Proof of Lemma 2

*Proof.* Let $\chi(S) \in \{0,1\}^d$ denote the indicator vector of subset $S \subseteq [d]$. Without loss of generality, we suppose that $a = \chi([k])$, $\hat{a} = \chi([s])$, and $a^* = \chi([s] \cup \{k\})$ for some $s \in \{0, 1, \ldots, k-1\}$. We then have $\hat{\ell}^\top a = \sum_{i=1}^k \ell_i$ and $\ell^{*\top} a = \sum_{i=1}^k \ell'_i$. It should be noted that $\hat{\ell}^\top a$ and $\ell^{*\top} a$ take values in $\{0, 1, \ldots, k\}$. Let us denote

$$P(i) = \mathrm{Prob}[\hat{\ell}^\top a = i], \quad P'(i) = \mathrm{Prob}[\ell^{*\top} a = i] \tag{11}$$

for $i = 0, 1, \ldots, k$. For $j = 1, 2, \ldots, k$, we denote $B_j = \sum_{i=1}^j b_i$ and $B'_j = \sum_{i=1}^j b'_i$, where $b_i$ and $b'_i$ stand for the values $b_i$ in (8) for generating $\ell^*$ and $\hat{\ell}$, respectively. Let us denote

$$Q_j(i) = \mathrm{Prob}[B_j = i], \quad Q'_j(i) = \mathrm{Prob}[B'_j = i] \tag{12}$$

for $i = 0, 1, \ldots, j$. Let us consider the conditional probability of $\ell^{*\top} a$ given $B_k$ and $u_0$ in (8). Given $B_k$ and $u_0 \in [0, \frac{1}{2}]$, $\ell^{*\top} a$ follows a uniform distribution over $\{B_k, B_k+1, \ldots, k\}$. Indeed, we obtain $\ell^*_j = 1$ for each $j \in I_\mathrm{p} := \{j' \in [k] \mid b_{j'} = 1\}$, as we have $u_j \geq \frac{1}{2} \geq u_0$. Because $u_j$ for $j \in I_\mathrm{n} := \{j' \in [k] \mid b_{j'} = 0\}$, and $u_0$ follows a uniform distribution over $[0, \frac{1}{2}]$ independently, $\sum_{j \in I_\mathrm{n}} \ell^*_j$, the number of $j \in I_\mathrm{n}$ with $u_j$ lager than $u_0$, follows a uniform distribution over $\{0, 1, \ldots, |I_\mathrm{n}|\}$. As we have $|I_\mathrm{p}| = B_k$ and $|I_\mathrm{n}| = k - B_k$, it holds that $\ell^{*\top} a = \sum_{j \in I_\mathrm{p}} \ell^*_j + \sum_{j \in I_n} \ell^*_j = B_k + \sum_{j \in I_n} \ell^*_j$ follows a uniform distribution over $\{B_k, B_k + 1, \ldots, k\}$, given $B_k$ and $u_0 \in [0, \frac{1}{2}]$. Similarly, given $B_k$ and $u_0 \in [\frac{1}{2}, 1]$, $\ell^{*\top} a$ follows a uniform distribution over $\{0, 1, \ldots, B_k\}$. Therefore, $P(i)$ can be expressed as

$$\begin{aligned}
P(i) &= Q_k(0)\frac{1}{2} \cdot \frac{1}{k+1} + Q_k(1)\frac{1}{2} \cdot \frac{1}{k} + \cdots + Q_k(i-1)\frac{1}{2} \cdot \frac{1}{k-i+2} + Q_k(i)\frac{1}{2} \cdot \frac{1}{k-i+1} \\
&\quad + Q_k(i)\frac{1}{2} \cdot \frac{1}{i+1} + Q_k(i+1)\frac{1}{2} \cdot \frac{1}{i+2} + \cdots + Q_k(k)\frac{1}{2} \cdot \frac{1}{k+1} \\
&= \frac{1}{2}\sum_{j=0}^i \frac{Q_k(j)}{k-j+1} + \frac{1}{2}\sum_{j=i}^k \frac{Q_k(j)}{j+1}
\end{aligned} \tag{13}$$

for each $i = 0, 1, \ldots, k$. Similarly, we obtain

$$P'(i) = \frac{1}{2}\sum_{j=0}^i \frac{Q'_k(j)}{k-j+1} + \frac{1}{2}\sum_{j=i}^k \frac{Q'_k(j)}{j+1}. \tag{14}$$

Therefore, we obtain

$$P(i) - P'(i) = \frac{1}{2}\sum_{j=0}^i \frac{Q_k(j) - Q'_k(j)}{k-j+1} + \frac{1}{2}\sum_{j=i}^k \frac{Q_k(j) - Q'_k(j)}{j+1}. \tag{15}$$

From the assumption that $\hat{a} = \chi([s])$, and $a^* = \chi([s] \cup \{k\})$ for $s \leq k-1$, we obtain $Q_{k-1}(j) = Q'_{k-1}(j)$ and $Q_k(j) = \frac{1}{2}Q_{k-1}(j) + \frac{1}{2}Q_{k-1}(j-1)$, $Q'_k(j) = (\frac{1}{2}+2\varepsilon)Q_{k-1}(j) + (\frac{1}{2}-2\varepsilon)Q_{k-1}(j-1)$, and therefore, we have $Q_k(j) - Q'_k(j) = 2\varepsilon(Q_{k-1}(j-1) - Q_{k-1}(j))$. By substituting this with (15), we obtain

$$\begin{aligned}
P(i) - P'(i) &= \varepsilon\sum_{j=0}^i \frac{Q_{k-1}(j-1) - Q_{k-1}(j)}{k-j+1} + \varepsilon\sum_{j=i}^k \frac{Q_{k-1}(j-1) - Q_{k-1}(j)}{j+1} \\
&= \varepsilon\left(\sum_{j=0}^{i-1} Q_{k-1}(j)\left(\frac{1}{k-j} - \frac{1}{k-j+1}\right) - \frac{Q_{k-1}(i)}{k-i+1} \right. \\
&\quad \left. + \frac{Q_{k-1}(i-1)}{i+1} + \sum_{j=i}^{k-1} Q_{k-1}(j)\left(\frac{1}{j+2} - \frac{1}{j+1}\right)\right) \\
&= \varepsilon\left(\frac{Q_{k-1}(i-1)}{i+1} - \frac{Q_{k-1}(i)}{k-i+1} + \sum_{j=0}^{i-1}\frac{Q_{k-1}(j)}{(k-j)(k-j+1)} - \sum_{j=i}^{k-1}\frac{Q_{k-1}(j)}{(j+1)(j+2)}\right).
\end{aligned}$$

The last term $\sum_{j=i}^{k-1} \frac{Q_{k-1}(j)}{(j+1)(j+2)}$ can be bounded as follows:

$$\sum_{j=i}^{k-1} \frac{Q_{k-1}(j)}{(j+1)(j+2)} \leq \sum_{j=0}^{k-1} \frac{Q_{k-1}(j)}{(j+1)(j+2)} \leq \mathrm{Prob}\left[B_{k-1} < \left\lfloor \frac{k}{4} \right\rfloor \right] + \sum_{j=\lfloor k/4 \rfloor}^{k-1} \frac{Q_{k-1}(j)}{(j+1)(j+2)}$$

$$\leq \mathrm{Prob}\left[ B_{k-1} - \mathbf{E}[B_{k-1}] \leq -\frac{k}{8} \right] + \sum_{j=\lfloor k/4 \rfloor}^{k-1} \frac{Q_{k-1}(j)}{(k/4)(k/4+1)} \leq \exp(-\frac{k}{32}) + \frac{16}{k^2} \leq \frac{2^{11}}{k^2},$$

where the third inequality comes from the fact that $\mathbf{E}[B_{t-1}] \geq \frac{3(k-1)}{8}$, and the fourth inequality comes from Hoeffding's inequality. In a similar way, we can demonstrate that $\sum_{j=0}^{i-1} \frac{Q_{k-1}(j)}{(k-j)(k-j+1)} \leq \frac{2^{11}}{k^2}$. Therefore, we obtain

$$|P(i) - P'(i)| \leq \varepsilon \left( \left| \frac{Q_{k-1}(i-1)}{i+1} - \frac{Q_{k-1}(i)}{k-i+1} \right| + \frac{2^{12}}{k^2} \right). \tag{16}$$

Next, we show that $\left| \frac{Q_{k-1}(i-1)}{i+1} - \frac{Q_{k-1}(i)}{k-i+1} \right| = O(1/k^2 + \varepsilon/k)$ by outlining that $R_j(i) := \frac{Q_j(i)}{Q_j(i-1)} \approx \frac{j+1-i}{i}$. Let us define $\beta = \frac{1+4\varepsilon}{1-4\varepsilon}$. We show that it holds for all $j = 1, 2, \ldots, k-1$ and $i = 1, \ldots, j$, such that

$$\frac{1}{\beta} \frac{j+1-i}{i} \leq R_j(i) \leq \frac{j+1-i}{i}, \tag{17}$$

by induction in $j$. For $j = 1$, (17) clearly holds. Because $Q_j$ corresponds to the probability distribution of $\sum_{i=1}^{j} b_i$, and $b_i$ for $k \leq k-1$ follows the Bernoulli distribution as defined in (8) with $a^* = \chi([s])$, $R_{j+1}$ can be expressed as follows:

$$R_{j+1}(i) = \frac{Q_{j+1}(i+1)}{Q_{j+1}(i)} = \frac{\alpha_j Q_j(i+1) + Q_j(i)}{\alpha_j Q_j(i) + Q_j(i-1)} = \frac{1 + \alpha_j R_j(i)}{\alpha_j + 1/R_j(i-1)} = \frac{1}{\alpha_j} \frac{1 + \alpha_j R_j(i)}{1 + 1/(\alpha_j R_j(i-1))}$$

for $j = 1, 2, \ldots, k-2$, where $\alpha_j = \beta$ for $j \leq s-1$ and $\alpha_j = 1$, otherwise. Assuming that (17) holds for $j = j'$ and $\alpha_{j'} = \beta$, we obtain

$$R_{j'+1}(i) = \frac{1}{\beta} \frac{1 + \beta R_{j'}(i)}{1 + 1/(\beta R_{j'}(i-1))} \leq \frac{1}{\beta} \frac{1 + \beta \frac{j'+1-i}{i}}{1 + \frac{1}{\beta} \frac{i-1}{j'+2-i}} = \frac{i + \beta(j'+1-i)}{i-1+\beta(j'+2-i)} \frac{j'+2-i}{i} \leq \frac{j'+2-i}{i}$$

$$R_{j'+1}(i) = \frac{1}{\beta} \frac{1 + \beta R_{j'}(i)}{1 + 1/(\beta R_{j'}(i-1))} \geq \frac{1}{\beta} \frac{1 + \frac{j'+1-i}{i}}{1 + \frac{i-1}{j'+2-i}} = \frac{1}{\beta} \frac{j'+2-i}{i},$$

which means that (17) also holds for incremented $j = j'+1$. Similarly in the case of $\alpha_{j'} = 1$, we can demonstrate that (17) for $j = j'$ implies that (17) holds for $j = j'+1$. Consequently, (17) holds for all $j \in \{1, 2, \ldots, k-1\}$ and $i \in \{0, 1, \ldots, i-1\}$. As a result, we obtain

$$\left| \frac{Q_{k-1}(i-1)}{i+1} - \frac{Q_{k-1}(i)}{k-i+1} \right| = Q_{k-1}(i-1) \left| \frac{1}{i+1} - \frac{R_{k-1}(i-1)}{k-i+1} \right|$$

$$\leq Q_{k-1}(i-1) \left( \left| \frac{1}{i+1} - \frac{k-i-1}{(i-1)(k-i+1)} \right| + \left| \left( 1 - \frac{1}{\beta} \right) \frac{k-i-1}{(i-1)(k-i+1)} \right| \right)$$

$$\leq Q_{k-1}(i-1) \left( \frac{2k}{(i+1)(i-1)(k-i+1)} + \frac{8\varepsilon}{i-1} \right)$$

From Hoeffding's inequality, it follows that for $i < \lfloor k/4 \rfloor$, $Q_{k-1}(i-1) \leq \exp(-k^2/32) \leq 2^{10}/k^2$. For $i \geq \lfloor k/4 \rfloor$, $\frac{2k}{(i+1)(i-1)(k-i+1)} \leq 2^{10}/k^2$. Therefore, the right-most-hand side above is bounded by $\frac{2^{10}}{k^2} + \frac{8\varepsilon Q_{k-1}(i-1)}{i-1}$. From this and (16), we obtain that

$$|P(i) - P'(i)| \leq \varepsilon \left( \frac{2^{13}}{k^2} + \frac{8\varepsilon Q_{k-1}(i-1)}{i-1} \right) \tag{18}$$

Further, from (13), we obtain

$$P(i) = \frac{1}{2} \sum_{j=0}^{i} \frac{Q_k(j)}{k-j+1} + \frac{1}{2} \sum_{j=i}^{k} \frac{Q_k(j)}{j+1} \geq \frac{1}{2} \sum_{j=0}^{k} \frac{Q_k(j)}{k+1} = \frac{1}{2(k+1)} \qquad (19)$$

From (18) and (19), $|P(i) - P'(i)|/P(i) \leq 1/2$ for $\varepsilon \leq 2^{-16}$. The KL divergence between $P$ and $P'$ is bounded as follows:

$$\begin{aligned}
\mathbf{KL}(\ell^{*\top} a \| \hat{\ell}^{\top} a) &= -\sum_{i=0}^{k} P(i) \log\left(\frac{P'(i)}{P(i)}\right) = -\sum_{i=0}^{k} P(i) \log\left(1 + \frac{P'(i) - P(i)}{P(i)}\right) \\
&\leq -\sum_{i=0}^{k} P(i) \left(\frac{P'(i) - P(i)}{P(i)} - 2\left(\frac{P'(i) - P(i)}{P(i)}\right)^2\right) \\
&\leq 2 \sum_{i=0}^{k} \frac{(P'(i) - P(i))^2}{P(i)} \\
&\leq 4(k+1)\varepsilon^2 \sum_{i=0}^{k} \left(\frac{2^{13}}{k^2} + \frac{8\varepsilon Q_{k-1}(i-1)}{i-1}\right)^2 \\
&\leq 8(k+1)\varepsilon^2 \left(\sum_{i=0}^{k} \left(\frac{2^{13}}{k^2}\right)^2 + \sum_{i=0}^{k} \left(\frac{8\varepsilon Q_{k-1}(i-1)}{i-1}\right)^2\right) \\
&\leq 8(k+1)\varepsilon^2 \left(\frac{2^{26}(k+1)}{k^4} + \frac{2^{16}\varepsilon^2}{k^{5/2}}\right) \leq \frac{2^{51}\varepsilon^2}{k^2} + \frac{2^{16}\varepsilon^4}{k^{3/2}},
\end{aligned}$$

where the first inequality comes from $\log(1+x) \geq x - 2x^2$ for $|x| \leq 1/2$; the second inequality comes from $\sum P(i) = \sum P'(i) = 1$; the third inequality comes from (18) and (19); the forth inequality comes from a standard inequality of $(x+y)^2 \leq 2(x^2 + y^2)$; and the fifth inequality comes from Hoeffding's inequality and $Q_{k-1}(i-1) \leq 2^{10} k^{-1/2}$. $\qquad \square$

# B   Proof of Lemma 3

We begin the proof with introducing notations: there is a one-to-one correspondence between $\mathcal{A}$ defined by (2) and the set of all injection $\sigma$ from $[k]$ to $[n]$. In fact, given an injection $\sigma : [k] \to [n]$, the indicator vector $\chi(S) \in \{0, 1\}^d$ of $S$ for $S = \{(i-1)n + j \mid i \in [k], j \in [n], \sigma(i) = j\}$ is an element of $\mathcal{A}$. Conversely, for any $a \in \mathcal{A}$, there is a unique injection $\sigma : [k] \to [n]$ such that $a_{(i-1)n+j} = 1$ if and only if $j = \sigma(i)$. Therefore, we can regard each element in $a \in \mathcal{A}$ as an injection from $[k]$ to $[n]$. For outputs $a_t$ from the algorithm, let us denote $\sigma_t : [k] \to [n]$ as the corresponding injection. Similarly, there is a one-to-one correspondence between $\mathcal{A}'$ and the set of all injection $\sigma$ from $[m]$ to $[n]$. Let $\sigma^* : [m] \to [n]$ denote the injection corresponding to $a^* \in \mathcal{A}'$. Then we have $a^* = \chi(\{(i-1)n + j \mid i \in [m], j \in [n], \sigma^*(i) = j\}) =: b(\sigma^*)$. Let $\Sigma_{m,n}$ denote the set of all injections from $[m]$ to $[n]$.

As we have $\mathbf{E}_{\ell_t \sim D_{a^*}}[\hat{\ell}_t^{\top} a] = \frac{k}{2} - \varepsilon \sum_{i=1}^{m} a_{(i-1)n + \sigma^*(i)}$ from (4), the expectation of the regret can be expressed as follows:

$$\mathbf{E}[R_T] = \sum_{t=1}^{T} \sum_{i=1}^{m} \varepsilon \left(1 - \mathbf{E}[a_{t,(i-1)n+\sigma^*(i)}]\right) = \varepsilon \left(mT - \sum_{i=1}^{m} \mathbf{E}\left[N_{(i-1)n+\sigma^*(i)}\right]\right), \qquad (20)$$

where the expectation is taken with respect to $\ell_t \sim D_{a^*} = D_{b(\sigma^*)}$ for $t \in [T]$. We consider bounding $\mathbf{E}\left[N_{(i-1)n+\sigma^*(i)}\right]$ by means of Lemma 1. For $\sigma^* : [k] \to [n]$, let us define $b'(\sigma^*) :=$

$\chi(\{(i-1)n+j \mid i \in [m-1], j \in [n], \sigma^*(i) = j\})$. From Lemma 1, we obtain

$$\left| \mathop{\mathbf{E}}_{\sigma^* \sim U(\Sigma_{m,n})} \mathop{\mathbf{E}}_{\ell_t \sim D_{b(\sigma^*)}} [N_{(m-1)n+\sigma(m)}] - \mathop{\mathbf{E}}_{\sigma^* \sim U(\Sigma_{m,n})} \mathop{\mathbf{E}}_{\ell_t \sim D_{b'(\sigma^*)}} [N_{(m-1)n+\sigma(m)}] \right|$$

$$\leq \mathop{\mathbf{E}}_{\sigma^* \sim U(\Sigma_{m,n})} \left| \mathop{\mathbf{E}}_{\ell_t \sim D_{b(\sigma^*)}} [N_{(m-1)n+\sigma(m)}] - \mathop{\mathbf{E}}_{\ell_t \sim D_{b'(\sigma^*)}} [N_{(m-1)n+\sigma(m)}] \right|$$

$$\leq T \mathop{\mathbf{E}}_{\sigma^* \sim U(\Sigma_{m,n})} \sqrt{\sum_{t=1}^{T} \mathop{\mathbf{E}}_{a_t \sim A_t(D_{b'(\sigma^*)})} \left[ \mathop{\mathbf{KL}}_{\ell \sim D_{b'(\sigma^*)}, \ell' \sim D_{b(\sigma^*)}} (a_t^\top \ell || a_t^\top \ell') \right]}$$

$$\leq T \sqrt{\mathop{\mathbf{E}}_{\sigma^* \sim U(\Sigma_{m,n})} \sum_{t=1}^{T} \mathop{\mathbf{E}}_{a_t \sim A_t(D_{b'(\sigma^*)})} \left[ \mathop{\mathbf{KL}}_{\ell \sim D_{b'(\sigma^*)}, \ell' \sim D_{b(\sigma^*)}} (a_t^\top \ell || a_t^\top \ell') \right]}, \tag{21}$$

where the first and the third inequalities follow from Jensen's inequality, and the second inequality follows from Lemma 1. Let us consider $\mathbf{KL}(a_t^\top \ell || a_t^\top \ell')$ for $\ell \sim D_{b'(\sigma^*)}$ and $\ell' \sim D_{b(\sigma^*)}$, having fixed $\sigma^* \in \Sigma_{m,n}$ and fixed $a_t \in \mathcal{A}$. If $a_{t,(m-1)n+\sigma(m)} = 1$, from the assumption of (7), we obtain $\mathbf{KL}(a_t^\top \ell || a_t^\top \ell') \leq C_D \varepsilon^2$. Otherwise, we have $\mathbf{KL}(a_t^\top \ell || a_t^\top \ell') = 0$ because the probabilistic distribution of $a_t^\top \ell$ is equal to that of $a_t^\top \ell'$. Consequently, we obtain

$$\sum_{t=1}^{T} \mathop{\mathbf{E}}_{a_t \sim A_t(D_{b'(\sigma^*)})} \left[ \mathop{\mathbf{KL}}_{\ell \sim D_{b'(\sigma^*)}, \ell' \sim D_{b(\sigma^*)}} (a_t^\top \ell || a_t^\top \ell') \right]$$

$$\leq \sum_{t=1}^{T} \mathop{\mathrm{Prob}}_{a_t \sim A_t(D_{b'(\sigma^*)})} [a_{t,(m-1)n+\sigma(m)} = 1] C_D \varepsilon^2 = \mathop{\mathbf{E}}_{\ell_t \sim D_{b'(\sigma^*)}} [N_{(m-1)n+\sigma(m)}] C_D \varepsilon^2 \tag{22}$$

Let us define $S \in \mathbb{R}$ by

$$S := \mathop{\mathbf{E}}_{\sigma^* \sim U(\Sigma_{m,n})} \mathop{\mathbf{E}}_{\ell_t \sim D_{b'(\sigma^*)}} [N_{(m-1)n+\sigma(m)}] \tag{23}$$

Combining the above two inequalities (21) and (22), we obtain

$$\mathop{\mathbf{E}}_{\sigma^* \sim U(\Sigma_{m,n})} \mathop{\mathbf{E}}_{\ell_t \sim D_{b(\sigma^*)}} [N_{(m-1)n+\sigma(m)}] \leq S + T\sqrt{C_D \varepsilon^2 S}. \tag{24}$$

Then we evaluate $S$ defined by (23). Let $\sigma^*|_{[m-1]}$ denote the restriction of $\sigma^* : [m] \to [n]$ to $[m-1]$, i.e., $\sigma^*|_{[m-1]} : [m-1] \to [n]$ is defined by $\sigma^*|_{[m-1]}(i) = \sigma^*(i)$ for $i \in [m-1]$. Let $R'(\sigma^*)$ denote the range of $\sigma^*|_{[m-1]}$. From the definition of $b'$, $b'(\sigma^*)$ does not depend on $\sigma^*$, but is determined by $\sigma^*|_{[m-1]}$. If $\sigma^*$ follows a uniform distribution over $\Sigma_{m,n}$, the posterior probability of $\sigma^*(m)$ given $\sigma^*|_{[m-1]}$ is a uniform distribution over $[n] \setminus R'(\sigma^*)$. Consequently, $S$ can be evaluated as follows:

$$S = \mathop{\mathbf{E}}_{\sigma^*|_{[m-1]}} \mathop{\mathbf{E}}_{\ell_t \sim D_{b'(\sigma^*)}} \mathop{\mathbf{E}}_{\sigma^*(m) \sim U([n] \setminus R'(\sigma^*))} [N_{(m-1)n+\sigma(m)}]$$

$$= \mathop{\mathbf{E}}_{\sigma^*|_{[m-1]}} \mathop{\mathbf{E}}_{\ell_t \sim D_{b'(\sigma^*)}} \left[ \frac{1}{|[n] \setminus R'(\sigma^*)|} \sum_{j \in [n] \setminus R'(\sigma^*)} N_{(m-1)n+j} \right]$$

$$= \mathop{\mathbf{E}}_{\sigma^*|_{[m-1]}} \mathop{\mathbf{E}}_{\ell_t \sim D_{b'(\sigma^*)}} \left[ \frac{1}{|[n] \setminus R'(\sigma^*)|} \sum_{t=1}^{T} \sum_{j \in [n] \setminus R'(\sigma^*)} a_{t,(m-1)n+j} \right] \leq \frac{T}{n-m+1}, \tag{25}$$

where the last inequality comes from $|R'(\sigma*)| = m - 1$, and from $a_t \in \mathcal{A}$ being defined by (2). Combining (24) and (25), we obtain

$$\mathop{\mathbf{E}}_{\sigma^* \sim U(\Sigma_{m,n})} \mathop{\mathbf{E}}_{\ell_t \sim D_{b(\sigma^*)}} [N_{(m-1)n+\sigma(m)}] \leq \frac{T}{n-m+1} + T\sqrt{\frac{C_D \varepsilon^2 T}{n-m+1}}.$$

As we assume $k \leq n$ and $n \geq 2$, we have $n - m + 1 = n - \lceil k/2 \rceil + 1 \geq n - \lceil n/2 \rceil + 1 \geq \max\{2, n/2\}$. Thereby, by setting $\varepsilon = \sqrt{\frac{n}{32 C_D T}}$, we obtain

$$\mathop{\mathbf{E}}_{\sigma^* \sim U(\Sigma_{m,n})} \mathop{\mathbf{E}}_{\ell_t \sim D_{b(\sigma^*)}} [N_{(m-1)n + \sigma(m)}] \leq \frac{T}{2} + T \sqrt{\frac{2 C_D \varepsilon^2 T}{n}} = \frac{T}{2} + \frac{T}{4} = \frac{3T}{4}.$$

For each $i \in [m]$ besides $m$, we can show $\mathbf{E}[N_{(i-1)n+\sigma(i)}] \leq \frac{3T}{4}$ in a similar way. Then, from this and (20), we obtain

$$\mathbf{E}[R_T] \geq \varepsilon \left( mT - \sum_{i=1}^{m} \frac{3T}{4} \right) = \frac{\varepsilon mT}{4} = \frac{m}{4} \sqrt{\frac{nT}{32 C_D}} \geq \frac{k}{8} \sqrt{\frac{nT}{32 C_D}} = \frac{1}{8} \sqrt{\frac{dkT}{32 C_D}}.$$

where the inequality comes from $m = \lceil k/2 \rceil \geq k/2$, and the last inequality comes from $d = kn$.

## C  Lower Bound for the Multiple-play Bandit Problem (Proof of Theorem 2)

For the multiple-play bandit problem, i.e., for $\mathcal{A} = \binom{[d]}{k}$, Observation 1 does not allow deriving directly a regret lower bound. In this subsection, we extend the observation to multiple-play bandit problems.

Let $U(\Sigma_d)$ denote a uniform distribution over all permutations of $[d]$. For a permutation $\sigma : [d] \to [d]$, let $\sigma([i])$ denote the element of $\{0,1\}^d$ such that the $\sigma(j)$-th component is 1 if $j \in [i]$, and 0, otherwise; i.e., $\sigma([i])$ is the indicator vector of $\{\sigma(j) \mid j \in [i]\}$. If $\sigma \sim U(\Sigma_d)$, then $\sigma([k])$ follows a uniform distribution $U(\mathcal{A})$ over $\mathcal{A} = \binom{[d]}{k}$. Consequently, we obtain $\mathop{\mathbf{E}}_{a^* \sim U(\mathcal{A})} \left[ \mathop{\mathbf{E}}_{\ell_1, \dots, \ell_T \sim D_{a^*}} [R_T] \right] = \mathop{\mathbf{E}}_{\sigma \sim U(\Sigma_d)} \left[ \mathop{\mathbf{E}}_{\ell_1, \dots, \ell_T \sim D_{\sigma([k])}} [R_T] \right]$. Let us define $M(k,i) \in \mathbb{R}$ by

$$M(k,i) = \mathop{\mathbf{E}}_{\sigma \sim U(\Sigma_d)} \left[ \mathop{\mathbf{E}}_{\ell_1, \dots, \ell_T \sim D_{\sigma([k])}} [N_{\sigma(i)}] \right].$$

Then, we obtain $M(k,1) = M(k,2) = \cdots = M(k,k)$ and $M(k,k+1) = M(k,k+2) = \cdots = M(k,d)$. From (5), the expectation of the regret can be expressed as follows:

$$\mathop{\mathbf{E}}_{\sigma \sim U(\Sigma_d)} \left[ \mathop{\mathbf{E}}_{\ell_1, \dots, \ell_T \sim D_{\sigma([k])}} [R_T] \right] = \varepsilon \left( kT - \sum_{i=1}^{k} M(k,i) \right) = \varepsilon(kT - kM(k,k)) = \varepsilon k(T - M(k,k)). \tag{26}$$

Let us evaluate $M(k,k)$ considering the difference between $M(k,k)$ and $M(k-1,k)$. From Lemma 1, we have

$$\left| \mathop{\mathbf{E}}_{\ell_1, \dots, \ell_T \sim D_{\sigma([k-1])}} [N_{\sigma(k)}] - \mathop{\mathbf{E}}_{\ell_1, \dots, \ell_T \sim D_{\sigma([k])}} [N_{\sigma(k)}] \right|$$
$$\leq T \sqrt{\sum_{t=1}^{T} \mathop{\mathbf{E}}_{a_t \sim A_t(D_{\sigma([k-1])})} \left[ \mathop{\mathbf{KL}}_{\ell \sim D_{\sigma([k-1])}, \ell' \sim D_{\sigma([k])}} (a_t^\top \ell \| a_t^\top \ell') \right]}. \tag{27}$$

Let us consider $\mathbf{KL}(a_t^\top \ell \| a_t^\top \ell')$ for $\ell \sim D_{\sigma([k-1])}$ and $\ell' \sim D_{\sigma([k])}$, having fixed $\sigma \in \Sigma_d$ and fixed $a_t \in \mathcal{A}$. If $a_{t,\sigma(k)} = 1$, we obtain $\mathbf{KL}(a_t^\top \ell \| a_t^\top \ell') = O(\frac{\varepsilon^2}{k^2} + \frac{\varepsilon^4}{k^{3/2}})$ from Lemma 2. Otherwise, we obtain $\mathbf{KL}(a_t^\top \ell \| a_t^\top \ell') = 0$, because the probabilistic distribution of $a_t^\top \ell$ is equal to that of $a_t^\top \ell'$. Therefore,

$$\sum_{t=1}^{T} \mathop{\mathbf{E}}_{a_t \sim A_t(D_{\sigma([k-1])})} \left[ \mathop{\mathbf{KL}}_{\ell \sim D_{\sigma([k-1])}, \ell' \sim D_{\sigma([k])}} (a_t^\top \ell \| a_t^\top \ell') \right]$$
$$\leq \sum_{t=1}^{T} \mathop{\mathrm{Prob}}_{a_t \sim A_t(D_{\sigma([k-1])})} [a_{t,\sigma(k)} = 1] \left( \frac{C\varepsilon}{k} \right)^2 = \mathop{\mathbf{E}}_{\ell_1, \dots, \ell_T \sim D_{\sigma([k-1])}} [N_{\sigma(k)}] \left( \frac{C\varepsilon}{k} \right)^2.$$

From this observation and (27), we obtain

$$\left| \mathop{\mathbf{E}}_{\ell_1,\ldots,\ell_T \sim D_{\sigma([k-1])}} [N_{\sigma(k)}] - \mathop{\mathbf{E}}_{\ell_1,\ldots,\ell_T \sim D_{\sigma([k])}} [N_{\sigma(k)}] \right| \le \frac{C\varepsilon T}{k} \sqrt{\mathop{\mathbf{E}}_{\ell_1,\ldots,\ell_T \sim D_{\sigma([k-1])}} [N_{\sigma(k)}]}.$$

Using this inequality, we obtain

$$|M(k-1,k) - M(k,k)| \le \mathop{\mathbf{E}}_{\sigma} \left[ \left| \mathop{\mathbf{E}}_{\ell_1,\ldots,\ell_T \sim D_{\sigma([k-1])}} [N_{\sigma(k)}] - \mathop{\mathbf{E}}_{\ell_1,\ldots,\ell_T \sim D_{\sigma([k])}} [N_{\sigma(k)}] \right| \right]$$

$$\le \frac{C\varepsilon T}{k} \mathop{\mathbf{E}}_{\sigma} \sqrt{\mathop{\mathbf{E}}_{\ell_1,\ldots,\ell_T \sim D_{\sigma([k-1])}} [N_{\sigma(k)}]} \le \frac{C\varepsilon T}{k} \sqrt{\mathop{\mathbf{E}}_{\sigma} \left[ \mathop{\mathbf{E}}_{\ell_1,\ldots,\ell_T \sim D_{\sigma([k-1])}} [N_{\sigma(k)}] \right]} = \frac{C\varepsilon T}{k} \sqrt{M(k-1,k)},$$

where the first and the last inequalities come from Jensen's inequality. Consequently, $M(k,k)$ is bounded as $M(k,k) \le M(k-1,k) + \frac{C\varepsilon T}{k}\sqrt{M(k-1,k)}$. Similarly, we can also show that $M(k,1) \le M(k-1,1) + \frac{C\varepsilon T}{k}\sqrt{M(k-1,1)}$. Considering that $M(k,1) = M(k,k)$, we obtain $M(k,k) \le \beta + \frac{C\varepsilon T}{k}\sqrt{\beta}$ for $\beta = \min\{M(k-1,k), M(k-1,1)\}$. As we have $M(k-1,1) = M(k-1,2) = \cdots = M(k-1,k-1)$ and $M(k-1,k) = \cdots = M(k-1,d)$, and $\sum_{i=1}^{d} M(k-1,i) = \mathop{\mathbf{E}}_{\sigma \sim U(\Sigma_d), \ell_1,\ldots,\ell_T \sim D_{\sigma([k-1])}} \left[ \sum_{i=1}^{d} N_i \right] = kT$, we have $\beta \le \frac{kT}{d}$. Therefore, we obtain

$$M(k,k) \le \frac{kT}{d} + \frac{C\varepsilon T}{k} \sqrt{\frac{kT}{d}} = T \left( \frac{k}{d} + C\sqrt{\frac{T}{dk}}\varepsilon \right). \text{ By setting } \varepsilon = 2^{-16} \min \left\{ k^{-1/4}, \frac{d-k}{2Cd}\sqrt{\frac{dk}{T}} \right\},$$

we obtain $M(k,k) \le T(\frac{k}{d} + \frac{d-k}{2d}) = \frac{T(d+k)}{2d}$. From (26), we have $\mathbf{E}[R_T] \ge \varepsilon k(T - M(k,k)) \ge \varepsilon kT \left( 1 - \frac{d+k}{2d} \right) = \frac{\varepsilon kT(d-k)}{2d} = 2^{-16} \min \left\{ \frac{1}{C} \left( \frac{d-k}{d} \right)^2 \sqrt{dk^3 T}, \quad \frac{d-k}{2d} k^{\frac{3}{4}} T \right\}. \quad \square$

## D  Upper Bounds

In this section, we provide the proof of Theorems 3. We consider the generalization of the bandit combinatorial optimization called *stochastic linear bandit* with the finite number of arms. In this problem, a player is given a finite decision set $\mathcal{A}$ before the game starts. In each round $t \in [T]$, the player chooses action $a_t \in \mathcal{A}$. Subsequently, observe loss $L_t = \ell^{*\top} a_t + \eta_t$, where $\eta_t$ is conditionally $\alpha$-subgaussian given $a_1, L_1, a_2, L_2, \ldots, a_{t-1}, L_{t-1}$ and $a_t$, i.e., $\mathbf{E}[\exp(\lambda \eta_t) \mid \mathcal{F}_t] \le \exp(\alpha^2 \lambda^2 / 2)$ almost surely, for $\mathcal{F}_t = \sigma(a_1, L_1, \ldots, a_{t-1}, L_{t-1}, a_t)$, which is the $\sigma$-algebra generated by $\{a_1, L_1, \ldots, a_{t-1}, L_{t-1}, a_t\}$. We suppose that the suboptimality gap $\max_{a,b \in \mathcal{A}} \ell^{*\top}(a-b)$ is at most $L$. Considering this problem, we define the regret $R_T'$ as follows:

$$R_T' = \max_{a \in \mathcal{A}} \sum_{t=1}^{T} \ell^{*\top}(a_t - a) = \sum_{t=1}^{T} \ell^{*\top} a_t - \min_{a \in \mathcal{A}} \sum_{t=1}^{T} \ell^{*\top} a = \sum_{t=1}^{T} \ell^{*\top}(a_t - a^*), \quad (28)$$

where we define that $a^* \in \arg\min_{a \in \mathcal{A}} \{\ell^{*\top} a\}$.

### D.1  Algorithm for Stochastic Linear Bandit for the Fixed Finite Number of Arms

We analyze $R_T'$ considering the output of Algorithm 1, which is obtained modifying Algorithm 12 in Section 22 of the preprint book by Lattimore and Szepesvári [19]. The differences between Algorithm 1 defined in the present study and Algorithm 12 in [19] are the following:

- In the related research, they deal only with the case in which the noise $\eta_t$ has a bounded variance, i.e., $\alpha = 1$. To deal with the case of general $\alpha$, we modify the definition (31) of $T_k$ in their algorithm.

- They assume that the suboptimality gap $\max_{a,b \in \mathcal{A}} \{\ell^{*\top}(a-b)\}$ is bounded by 1. To cope with changing suboptimality gaps, we modify the definition of $\varepsilon_t$ in their algorithm.

- They basically consider maximization problems, while we consider minimization (it does not result in essential differences).

Algorithm 1 is controlled by parameters $\varepsilon_1 > 0$ and $\delta > 0$. The algorithm divides rounds into *phases*: the $k$-th phase consists of $T_k$ rounds, where $T_k$ will be defined later. In each phase, a subset $\mathcal{A}_k$ of action set $\mathcal{A}$ is maintained. The algorithm chooses actions from $\mathcal{A}_k$ in the $k$-th phase, and $\mathcal{A}_k$ does not change over all rounds in this phase. At the beginning of each phase, the algorithm constructs a probabilistic measure $\pi_k$ over $\mathcal{A}_k$ satisfying the following condition:

$$\max_{b \in \mathcal{A}_k} \left\{ b^\top \left( \sum_{a \in \mathcal{A}_k} \pi_k(a) aa^\top \right)^{-1} b \right\} \leq d, \quad \|\pi_k\|_0 \leq \frac{d(d+1)}{2} + 1. \tag{29}$$

This measure always exists for all $k$. Indeed, as showed by Kiefer and Wolfowitz [15], if $\mathrm{span}(\mathcal{A}) = \mathbb{R}^d$, a maximizer $\pi^*$ of $\det(\sum_{a \in \mathcal{A}} \pi(a) aa^\top)$ satisfies the following condition:

$$\max_{b \in \mathcal{A}} \left\{ b^\top \left( \sum_{a \in \mathcal{A}} \pi^*(a) aa^\top \right)^{-1} b \right\} = d. \tag{30}$$

Even if $\mathrm{span}(\mathcal{A})$ is not equal to $\mathbb{R}^d$, equivalently, if $d' = \dim(\mathrm{span}(\mathcal{A}))$ is smaller than $d$, we can obtain $\pi^*$ for which the left-hand side of (30) is equal to $d'$, by maximizing $\det(\sum_{a \in \mathcal{A}} \pi(a)(Ba)(Ba)^\top)$ for an appropriate matrix $B \in \mathbb{R}^{d' \times d}$. Consequently, the left inequality of (29) can be satisfied. Further, Carathéodory's theorem implies that for arbitrary $\pi \in \Delta^{\mathcal{A}} := \{\pi : \mathcal{A} \to \mathbb{R}_{\geq 0} \mid \sum_{a \in \mathcal{A}} \pi(a) = 1\}$, there exists $\pi' \in \Delta^{\mathcal{A}}$ such that

$$\sum_{a \in \mathcal{A}} \pi(a) aa^\top = \sum_{a \in \mathcal{A}} \pi'(a) aa^\top, \quad \|\pi'\|_0 \leq \dim(\mathrm{span}\{aa^\top \mid a \in \mathcal{A}\}) + 1.$$

The dimensionality of $\mathrm{span}\{aa^\top \mid a \in \mathcal{A}\}$ is at most $d(d+1)/2$, which is the dimensionality of the linear space of all symmetric matrices of size $d$. Consequently, the right inequality of (29) can be satisfied.

The $k$-th phase consists of $T_k$ rounds from the $(t_k + 1)$-th round to the $t_{k+1}$-th round, in which the algorithm chooses action $a \in \mathcal{A}_k$ in exactly $T_k(a)$ rounds for each $a \in \mathcal{A}_k$. Here, $T_k(a)$ $(a \in \mathcal{A}_k)$, $T_k$, and $t_k$ are defined as follows:

$$T_k(a) = \left\lceil \frac{2d\alpha^2 \pi_k(a)}{\varepsilon_k^2} \log \frac{2|\mathcal{A}|k(k+1)}{\delta} \right\rceil, \quad T_k = \sum_{a \in \mathcal{A}_k} T_k(a), \quad t_k = \sum_{j=1}^{k-1} T_j, \tag{31}$$

where $\varepsilon_k = 2^{-k+1}\varepsilon_1$. At the end of the $k$-th phase, the algorithm calculates the least squares estimator $\hat{\ell}_k$ of $\ell^*$ by

$$\hat{\ell}_k = V_k^{-1} \sum_{t=t_k+1}^{t_k+T_k} r_t a_t \quad \text{with} \quad V_k = \sum_{a \in \mathcal{A}_k} T_k(a) aa^\top = \sum_{t=t_k+1}^{t_k+T_k} a_t a_t^\top. \tag{32}$$

Moreover, $\mathcal{A}_{k+1}$ is defined by eliminating actions that are not promising as follows:

$$\mathcal{A}_{k+1} = \left\{ a \in \mathcal{A}_k \; \middle| \; \min_{b \in \mathcal{A}_k} \hat{\ell}_k^\top (b - a) \geq -2\varepsilon_k \right\}. \tag{33}$$

The outputs of Algorithm 1 correspond to the regret upper bound in Theorem 4. The proof is provided in Appendix E.

## D.2 Proof of Theorem 3

In this subsection, we prove Theorem 3 by means of Theorem 4. Suppose that $\ell_t$ follows an arm-wise independent distribution $D^*$, i.i.d. Then, the bandit combinatorial optimization for $\{\ell_t\}$ is a special case of the stochastic linear bandits with $\ell^* = \mathop{\mathbf{E}}_{\ell \sim D^*}[\ell]$ and $\eta_t = (\ell_t - \eta^*)^\top a_t$. In this problem, the suboptimality gap $L = \max_{a,b \in \mathcal{A}} \ell^{*\top}(a - b)$ is at most $k$ because $\ell^* \in [0, 1]$ and $\mathcal{A} \subseteq \{a \in \{0, 1\}^d \mid \|a\|_0 = k\}$. Further, $\eta_t$ is $\sqrt{k}/2$-subgaussian from Hoeffding's Lemma, as

---

**Algorithm 1** Algorithm for stochastic linear bandits with the finite arms

---
**Require:** $\mathcal{A} \subseteq \mathbb{R}^d, \alpha, \delta, \varepsilon_1$
1: Set $\mathcal{A}_1 = \mathcal{A}, t_1 = 0$.
2: **for** $k = 1, \ldots$ **do**
3:     Let $\pi_k \in \Delta^{\mathcal{A}_k}$ be a probabilistic measure over $\mathcal{A}_k$ such that (29) is satisfied.
4:     Define $T_k(a), T_k$ and $t_{k+1}$ by (31)
5:     Choose action $a \in \mathcal{A}_k$ exactly $T_k(a)$ times, from the $(t_k + 1)$-th round to the $t_{k+1}$-th round.
6:     Calculate empirical estimate $\hat{\ell}_k$ of $\ell^*$ by (32).
7:     Eliminate arms with a high estimated loss based on (33).
8: **end for**

---

$\ell_{t1}, \ell_{t2}, \ldots, \ell_{td}$ are independent and $[0, 1]$-valued random variables. Thereby, applying Algorithm 1 with $L = k, \alpha = \sqrt{k}/2$ and $\delta = \delta'/2$, we obtain

$$R'_T = \sum_{t=1}^{T} \ell^{*\top}(a_t - a^*) \leq 64\sqrt{dkT \log \frac{2|\mathcal{A}| \log T}{\delta'}} + \frac{dk^2}{\varepsilon_1^2} \log \frac{2|\mathcal{A}|}{\delta'} + (k + 16\varepsilon_1)d^2 \quad (34)$$

with probability $1 - \delta'/2$. Moreover, we obtain $R_T - R'_T = O(\sqrt{kT \log(|\mathcal{A}|/\delta')})$ with probability $1 - \delta'/2$. In fact, we have $\sum_{t=1}^{T} \ell_t^\top (a_t - a) - R'_T \leq \sum_{t=1}^{T} \eta_t - \sum_{t=1}^{T} (\ell_t - \ell^*)^\top a$ for all $a \in \mathcal{A}$, and from the Azuma-Hoeffding inequality, we have

$$\mathrm{Prob} \left[ \sum_{t=1}^{T} \eta_t \geq \sqrt{\frac{kT}{2} \log \frac{4|\mathcal{A}|}{\delta'}} \right] \leq \frac{\delta'}{4|\mathcal{A}|}, \quad \mathrm{Prob} \left[ \sum_{t=1}^{T} (\ell^* - \ell_t)^\top a \geq \sqrt{\frac{kT}{2} \log \frac{4|\mathcal{A}|}{\delta'}} \right] \leq \frac{\delta'}{4|\mathcal{A}|}.$$

From the second inequality, the probability that there exists $a \in \mathcal{A}$ such that $\sum_{t=1}^{T} (\ell^* - \ell_t)^\top a \geq \sqrt{\frac{kT}{2} \log \frac{4|\mathcal{A}|}{\delta'}}$ is at most $\delta'/4$. Combining this observation and the aforementioned first inequality, we obtain $R_T - R'_T \leq \sqrt{2kT \log \frac{4|\mathcal{A}|}{\delta'}}$ with probability $1 - \delta'/2$. From this statement and (34), with probability $1 - \delta'$, we have $R_T \leq R'_T + \sqrt{2kT \log \frac{4|\mathcal{A}|}{\delta'}} \leq 66\sqrt{dkT \log \frac{2|\mathcal{A}| \log T}{\delta'}} + \frac{dk^2}{\varepsilon_1^2} \log \frac{2|\mathcal{A}|}{\delta'} + (k + 16\varepsilon_1)d^2 \leq 66\sqrt{dk^2T \log \frac{2ed \log T}{k\delta'}} + \frac{dk^3}{\varepsilon_1^2} \log \frac{2ed}{k\delta'} + (k + 16\varepsilon_1)d^2$. By setting $\varepsilon_1 = \Theta(k)$ and $\delta' = \Theta(\sqrt{d}T)$, we obtain $R_T = O(\sqrt{dk^2T \log(ed \log T/k\delta')} + d^2k)$ with probability $1 - \delta'$, and $R_T = O(kT)$ with probability $\delta'$. Consequently, we obtain $\mathbf{E}[R_T] = O(\sqrt{dk^2T \log(ed \log T/k\delta')} + d^2k + \delta'kT) = O(\sqrt{dk^2T \log T \log(ed/k)})$ for $T = \Omega(d^3)$. $\quad \square$

# E   Analysis of the Regret for Algorithm 1

In this section, we provide a proof of Theorems 4. To derive an upper bound of the regret defined by (28), we first consider a confidence bound for $\hat{\ell}_k^\top a$. From the standard analysis of confidence bounds for least squares estimators (see, e.g., [19]), for all $b \in \mathbb{R}^d$, we have

$$\mathrm{Prob} \left[ |(\hat{\ell}_k - \ell^*)^\top b| \geq \alpha \sqrt{2b^\top V_k^{-1} b \log \frac{2|\mathcal{A}|k(k+1)}{\delta}} \,\middle|\, \mathcal{F}_{t_{k-1}} \right]$$
$$\leq \exp\left( -\log \frac{2|\mathcal{A}|k(k+1)}{\delta} \right) = \frac{\delta}{|\mathcal{A}|k(k+1)}. \quad (35)$$

From the definitions of $V_k, T_k$ and $\pi_k$, it holds for all $b \in \mathcal{A}_k$ that

$$b^\top V_k^{-1} b = b^\top \left( \sum_{a \in \mathcal{A}_k} T_k(a)aa^\top \right)^{-1} b$$

$$\leq \left( \frac{2d\alpha^2}{\varepsilon_k^2} \log \frac{2|\mathcal{A}|k(k+1)}{\delta} \right)^{-1} b^\top \left( \sum_{a \in \mathcal{A}_k} \pi_k(a)aa^\top \right)^{-1} b \leq \left( \frac{2\alpha^2}{\varepsilon_k^2} \log \frac{2|\mathcal{A}|k(k+1)}{\delta} \right)^{-1},$$

where the first equality comes from (32), and the first and the second inequalities come from (31) and (29), respectively. Combining the above and (35), we obtain

$$\mathrm{Prob}[|(\ell^* - \hat{\ell}_k)^\top b| \geq \varepsilon_k | \mathcal{F}_{t_{k-1}}] \leq \frac{\delta}{|\mathcal{A}|k(k+1)}$$

for all $k$ and $b \in \mathcal{A}_k$. Therefore, we obtain

$$\mathrm{Prob}\left[\exists k, \ \exists b \in A_k, \ |(\ell^* - \hat{\ell}_k)^\top b| > \varepsilon_k\right] \leq \sum_{k=1}^{\infty} \frac{|\mathcal{A}_k|\delta}{|\mathcal{A}|k(k+1)} \leq \delta \sum_{k=1}^{\infty} \frac{1}{k(k+1)} = \delta.$$

In the discussion, hereinafter, we assume that

$$|(\ell^* - \hat{\ell}_k)^\top a| \leq \varepsilon_k \text{ for all } k \in \{1, 2, \ldots\} \text{ and } a \in \mathcal{A}_k. \tag{36}$$

Then, for all $k = 1, 2, \ldots$, we have $a^* \in \mathcal{A}_k$ because $a^* \in \mathcal{A}_1 = \mathcal{A}$ and $\hat{\ell}_k^\top(b - a^*) \geq \ell^{*\top}(b - a^*) - 2\varepsilon_k \geq -2\varepsilon$ for all $k$. Further, we obtain

$$\ell^{*\top}(a - a^*) \leq 8\varepsilon_k \text{ for all } k \in \{2, 3, 4, \ldots\} \text{ and } a \in \mathcal{A}_k. \tag{37}$$

Indeed, if $\ell^{*\top}(a - a^*) > 8\varepsilon_k$, we can see that $a \notin \mathcal{A}_k$ in both cases of (i) $a \notin \mathcal{A}_{k-1}$ and $a \in \mathcal{A}_{k-1}$: (i)as $\mathcal{A}_k \subseteq \mathcal{A}_{k-1}$ from the definition (33) of $\mathcal{A}_k$, $a \notin \mathcal{A}_{k-1}$ implies $a \notin \mathcal{A}_k$; (ii)assuming $a, a^* \in \mathcal{A}_{k-1}$, $\ell^{*\top}(a - a^*) > 8\varepsilon_k = 4\varepsilon_{k-1}$ and (36), we obtain $\hat{\ell}_{k-1}^\top(b - a) \geq \ell^{*\top}(b-a) - 2\varepsilon_{k-1} \geq \ell^{*\top}(a^* - a) - 2\varepsilon_{k-1} > 2\varepsilon_{k-1}$ for all $b \in \mathcal{A}_{k-1}$, which implies $a \notin \mathcal{A}_k$ from the definition (33) of $\mathcal{A}_k$.

Let us define $k(t)$ to be $k \in \{1, 2, \ldots\}$ such that $t_k < t \leq t_{k+1}$. Because $t_k < t \leq t_{k+1}$ means $a_t \in \mathcal{A}_k$, we have $a_t \in \mathcal{A}_{k(t)}$ for all $t$. Consequently, from (37), we have $\ell^{*\top}(a_t - a^*) \leq 8\varepsilon_{k(t)}$ for $t > T_1$. Therefore, for $T \geq T_1$, we have

$$\sum_{t=1}^{T} \ell^{*\top}(a_t - a^*) \leq LT_1 + \sum_{t=t_2}^{T} \ell^{*\top}(a_t - a^*) \leq LT_1 + 8\sum_{t=t_2}^{T} \varepsilon_{k(t)}$$

$$\leq LT_1 + 8\sum_{t=t_2}^{t_{k(T)+1}} \varepsilon_{k(t)} \leq LT_1 + 8\sum_{k=2}^{k(T)} T_k\varepsilon_k. \tag{38}$$

Let us evaluate $T_k$ and $t_k = \sum_{j=1}^{k-1} T_j$. From the definition (31) of $T_k$, we have

$$\frac{2d\alpha^2}{\varepsilon_k^2} \log \frac{2|\mathcal{A}|k(k+1)}{\delta} \leq T_k < \frac{2d\alpha^2}{\varepsilon_k^2} \log \frac{2|\mathcal{A}|k(k+1)}{\delta} + \frac{d(d+1)}{2} + 1, \tag{39}$$

as $T_k(a) - \frac{2d\alpha^2 \pi_k(a)}{\varepsilon_k^2} \log \frac{2|\mathcal{A}|k(k+1)}{\delta} \in [0, 1)$ for at most $d(d+1)/2+1$ actions $a \in \mathcal{A}$ and $T_k(a) = 0$ for the other actions. From the left inequality of (39), for $T > T_1$, we have

$$T > t_{k(T)} \geq T_{k(T)-1} \geq \frac{2d\alpha^2}{\varepsilon_{k(T)-1}^2} \log \frac{2|\mathcal{A}|k(T)(k(T)-1)}{\delta} \geq \frac{d\alpha^2 2^{2k(T)}}{8\varepsilon_1^2} \log \frac{2|\mathcal{A}|k(T)}{\delta},$$

which implies that

$$2^{k(T)} \leq \frac{\varepsilon_1}{\alpha} \sqrt{\frac{8T}{d}} \left(\log \frac{2|\mathcal{A}|k(T)}{\delta}\right)^{-1/2}, \quad k(T) \leq \log_2\left(\frac{\varepsilon_1}{\alpha} \sqrt{\frac{8T}{d}}\right) \leq \log T. \tag{40}$$

From this inequality and (37), we have

$$\sum_{m=2}^{k(T)} T_m \varepsilon_m \leq \sum_{m=2}^{k(T)} \left( \frac{2d\alpha^2}{\varepsilon_m} \log \frac{2|\mathcal{A}|m(m+1)}{\delta} + \frac{\varepsilon_m(d+2)^2}{2} \right)$$

$$\leq \sum_{m=2}^{k(T)} \left( \frac{2d\alpha^2}{\varepsilon_1 2^{-m+1}} \log \frac{2|\mathcal{A}|k(T)(k(T)+1)}{\delta} + \frac{\varepsilon_1 2^{-m+1}(d+2)^2}{2} \right)$$

$$\leq \frac{2d\alpha^2 2^{k(T)}}{\varepsilon_1} \log \frac{2|\mathcal{A}|k(T)(k(T)+1)}{\delta} + 2\varepsilon_1 d^2$$

$$\leq 2\alpha\sqrt{8dT} \left( \log \frac{2|\mathcal{A}|k(T)}{\delta} \right)^{-1/2} \cdot 2 \log \frac{2|\mathcal{A}|k(T)}{\delta} + 2\varepsilon_1 d^2$$

$$\leq 4\alpha\sqrt{8dT \log \frac{2|\mathcal{A}|k(T)}{\delta}} + 2\varepsilon_1 d^2 \leq 16\alpha\sqrt{dT \log \frac{|\mathcal{A}|k(T)}{\delta}} + 2\varepsilon_1 d^2,$$

where the first inequality comes from the right inequality of (39), the fourth inequality comes from the left inequality of (39) and the fact that $2|\mathcal{A}|k(T)(k(T)+1) \leq (2|\mathcal{A}|k(T))^2$. From the above inequality and (38), we obtain

$$\sum_{t=1}^{T} \ell^{*\top}(a_t - a^*) \leq LT_1 + 128\alpha\sqrt{dT \log \frac{|\mathcal{A}| \log T}{\delta}} + 16\varepsilon_1 d^2$$

$$\leq 128\alpha\sqrt{dT \log \frac{|\mathcal{A}| \log T}{\delta}} + \frac{4dL\alpha^2}{\varepsilon_1^2} \log \frac{|\mathcal{A}|}{\delta} + (L + 16\varepsilon_1)d^2.$$

□