[Reviews · NeurIPS 2019]

Reviewer 1



This work investigates combinatorial bandits (where actions are a subset of the corners of the boolean hypercube) both in the nonstochastic and stochastic settings, where actions are assumed to have exactly k bits equal to 1. In the nonstochastic setting, the lower bound recently proved by Cohen et al. is improved on by removing a spurious 1/sqrt{log T} factor, thus obtaining a tight regret bound of order sqrt{k^3dT}. This is achieved by generating correlated binary loss vectors, in contract to the approach followed by Cohen et al. in which correlated Gaussian losses were used. In the stochastic setting, the authors prove an upper bound of sqrt{k^2dT} (excluding log factors) for the case of losses with independent components in [0,1]. The paper is well written and the technical parts are generally clear. The trick used to shave the 1/sqrt{log T} factor is simple and elegant. The techniques of Cohen et al. are extended to handle sums of correlated binary random variables. The upper bound for the stochastic case appears to be a simple extension of Lattimore and Szepesvari. The contribution is solid and technically sound. However, its significance is unclear because neither the results nor the techniques are particularly surprising in the context of previous works. AFTER AUTHOR FEEDBACK I appreciated the feedback provided by the authors to my comments. However, I will stand by my score which I think is the most appropriate in the context of NeurIPS.

Reviewer 2



The paper has several new results on bandit combinatorial optimization. There are a few different ideas, but the main unifying theme is a new construction of hard instances for bandit combinatorial optimization. Mainly the paper is refining the results of Cohen, Hazan, and Koren (https://arxiv.org/pdf/1702.07539.pdf). First, this construction is used to refine a previous lower bound of Cohen, Hazan, and Koren for general bandit combinatorial optimization which used Gaussian noise to construct a hard instance. The authors turn it into a binary {0,1} distribution. This allows the authors to save a sqrt(log(T)) factor in the general bandit combinatorial optimization problem, since Gaussian noise must be divided by a logarithmic factor to almost surely stay bounded. The resulting lower bound is tight up to a T-independent log(d/k) factor for general bandit combinatorial optimization. The authors also prove that bandit combinatorial optimization becomes easier by a factor of sqrt(k) - for k the number of arms picked per round - with independent arms. The main point is to construct a distribution over actions ensuring a low variance observation whenever the losses of the coordinates are independent. This can be done by a direct use of the Kiefer-Wolfowitz theorem. Finally, there is an improved lower bound for online max matching in the symmetric K_{n,n} case. The previous Alon/Hazan/Koren work gave a lower bound which applied for K_{n,k} with k

Reviewer 3



[Post-response comment: The response addressed most of my comments. In particular, the gap in the analysis is due to my mis-reading the formula, and the response convinced me. However, the paper overall looks incremental, so it is a paper nice to have, but its acceptance seems to be depending on the quality of other papers.] The paper studies the bandit combinatorial optimization problem and improve the lower bound of the problem from $\Omega(\sqrt{dk^3T/log T})$ in the prior work [8] to $\Omega(\sqrt{dk^3T})$, removing a factor of $1/\sqrt{\log T}$. This makes the regret dependency on T and k, d tight up to a logarithmic factor. The analysis is built upon prior work [2,8], with the major innovation being a design of new distribution of loss vectors (given in Eq.(8)) that leads to a better lower bound. The design of distribution of the loss vectors looks nontrivial to me, and there are some involved technical analysis to make it work. However, there seems to be a gap in the analysis: In the derivation sequence between line 248 and line 249, I do not understand the last equality (after the inequality). It does not look to be true to me, since it is true only when the first time of the LHS of the equation, (P'(i)-P(i))/P(i) is set to zero, but this is not the case. With this gap, I am not sure how the rest derivation should proceed, and thus I am left with some doubt on the correctness of the analysis, although I feel that it should be amendable. I would be happy to raise my score if the authors could clarify this issue and fill this gap in the analysis. Beside this gap, I feel that the authors in general did a good job in explaining the problem, and in explaining the inuition and the idea of the analysis. The results look nice, in tighting up a regret bound, and in proving a potential useful distribution of loss vectors for future analysis. Therefore, I feel that the paper is worth to be published.

[Author Response · NeurIPS 2019]

Dear Reviewer #1:

> the result is far from being unexpected.

We believe that some readers may find unexpected results in our work. For example, the authors of the literature [8]

asked the following question in their conclusion: "For the problem of online ranking,... In particular, is the optimal

regret $\Theta(n^2\sqrt{T})$ in this setting?". To this question, our Theorem 1 gives the following answer: "No, the optimal regret

is $\Theta(n^{5/2}\sqrt{T})$".

> its significance is unclear because neither the results nor the techniques are particularly surprising in the

context of previous works.

We consider that our analysis includes nontrivial techniques, especially in the construction of hard instances. As

presented in lines 110-118, we carefully control the parameters of distributions to obtain desired lower bounds.

Dear Reviewer #2:

> I think the writing could be improved. In particular, I found reading this paper confusing because there were

several different results, and the introduction kept skipping between results and proof ideas/intuition and

descriptions of new problems and so on. I think for this paper it might be better to first clearly list out all

the problem types that will be considered, then list out all the new results, and then finally say a bit of proof

summary/preview.

Thanks for your suggestion. We will modify the manuscript so that all results and techniques are clarified.

> Theorem 2 says "mutiple"

Thanks for pointing out the typo. We will fix it in the revised version.

> The definition of $\hat{\ell}$ isn't really given, it's just implicit that $\hat{\ell}$ corresponds to $\hat{a}$. Better to be explicit.

In (8), we define a *class* of distributions $D_{a^*}$ *for all* $a^* \in \{0,1\}^d$, and $\hat{\ell}$ is defined to be a random vector following the

distribution $D_{\hat{a}}$ given by (8). We will describe this explicitly in the revised manuscript. Thanks for your suggestion.

> As the authors suggest, it would be great to get the correct rate up to an absolute constant.

We are now tackling this problem, but have not found the answer yet. We believe that our lower bound is tight and that

one can improve the upper bound to find the correct rate. To achieve this, however, a novel idea seems to be needed as

the upper bound has not been improved since 2009.

Dear Reviewer #3:

    > there seems to be a gap in the analysis: In the derivation sequence between line 248 and line 249, I do not
understand the last equality (after the inequality). It does not look to be true to me, since it is true only when
the first time of the LHS of the equation, (P'(i)-P(i))/P(i) is set to zero, but this is not the case.
> I expect the authors could fill the gap I mentioned above, and explain how the analysis should proceed,
then I would be happy to raise my score.

    Our analysis proceeds as follows:

$$-\sum_{i=0}^{k} P(i)\left(\frac{P'(i)-P(i)}{P(i)} - 2\left(\frac{P'(i)-P(i)}{P(i)}\right)^2\right) = -\sum_{i=0}^{k} P(i)\frac{P'(i)-P(i)}{P(i)} + 2\sum_{i=0}^{k} P(i)\left(\frac{P'(i)-P(i)}{P(i)}\right)^2$$

$$= -\sum_{i=0}^{k}(P'(i)-P(i)) + 2\sum_{i=0}^{k}\frac{(P'(i)-P(i))^2}{P(i)} = -\sum_{i=0}^{k}P'(i) + \sum_{i=0}^{k}P(i) + 2\sum_{i=0}^{k}\frac{(P'(i)-P(i))^2}{P(i)}.$$

Since $P$ and $P'$ are probability mass functions over $\{0, 1, \ldots, k\}$, we have $\sum_{i=0}^{k} P(i) = \sum_{i=0}^{k} P'(i) = 1$. Hence,

we have $-\sum_{i=0}^{k} P'(i) + \sum_{i=0}^{k} P(i) = -1 + 1 = 0$. By substituting this to the above displayed equation, we obtain

$-\sum_{i=0}^{k} P(i)\left(\frac{P'(i)-P(i)}{P(i)} - 2\left(\frac{P'(i)-P(i)}{P(i)}\right)^2\right) = 2\sum_{i=0}^{k}\frac{(P'(i)-P(i))^2}{P(i)}$. We hope the above discussion fills the gap

you mentioned. In the revised version, we will clarify how this equality is derived.

[Meta-Review · NeurIPS 2019]

Thanks for the clarification on the proof. The reviewers all agree that this is a fine result.